nature
ecology & evolution

# Allorecognition genes drive reproductive isolation in *Podospora anserina*

S. Lorena Ament-Velásquez [1,2 ✉], Aaron A. Vogan [1], Alexandra Granger-Farbos[3],
Eric Bastiaans [1,4], Ivain Martinossi-Allibert[1,5], Sven J. Saupe[3], Suzette de Groot[4], Martin Lascoux [6],
Alfons J. M. Debets [4], Corinne Clavé[3] and Hanna Johannesson [1 ✉]

**Allorecognition, the capacity to discriminate self from conspecific non-self, is a ubiquitous organismal feature typically governed by genes evolving under balancing selection. Here, we show that in the fungus *Podospora anserina*, allorecognition loci controlling vegetative incompatibility (*het* genes), define two reproductively isolated groups through pleiotropic effects on sexual compatibility. These two groups emerge from the antagonistic interactions of the unlinked loci *het-r* (encoding a NOD-like receptor) and *het-v* (encoding a methyltransferase and an MLKL/HeLo domain protein). Using a combination of genetic and ecological data, supported by simulations, we provide a concrete and molecularly defined example whereby the origin and coexistence of reproductively isolated groups in sympatry is driven by pleiotropic genes under balancing selection.**

The capacity to discriminate self from non-self occurs throughout the Tree of Life[1–3] and is necessary for fundamental processes such as multicellular growth, detection of pathogens and choice of mating partners. Genetically, the detection of non-self is achieved by the product of highly polymorphic genes that are subject to various forms of balancing selection, in particular negative frequency dependence where rare alleles have a fitness advantage[4–8]. Since balancing selection ensures the coexistence of several variants in a population, it can be hypothesized that self/non-self recognition genes with pleiotropic interactions on sexual reproduction could lead to the evolution of reproductively isolated groups and hence speciation. For instance, the major histocompatibility complex (MHC), which is an essential element of the adaptive immune system in jawed vertebrates, appears to contribute to and accelerate speciation by favouring assortative mating[9–11]. Likewise, in organisms such as fungi, slime moulds and colonial marine invertebrates, vegetative incompatibility (allorecognition) systems might lead to reproductive isolation if they display pleiotropic effects on sexual reproduction.

Here, we address this hypothesis directly using the model fungus *Podospora anserina*, for which the allorecognition system is well-studied[12–14]. As in other fungi, vegetative fusion between individuals of *P. anserina* is controlled by the so-called *het* (for heterokaryon incompatibility) genes. Fusion within a fungal mycelium is generally regarded as advantageous for cytoplasmic transport and colony establishment[15,16]. However, fusion between different individuals can be deleterious since it allows for the transmission of viruses, defective plasmids and selfish nuclei[17–21]. It is believed that the primary function of the *het* genes is to avoid such risks, as successful fusion is only possible if individuals are compatible at all of their *het* genes, otherwise triggering regulated cell death of the fused cell[22]. It has also been proposed that some *het* genes are involved in pathogen recognition and that the vegetative

incompatibility is a secondary by-product of their evolution[23,24]. In that sense, the *het* genes can be seen as analogous to components of the innate immune system of animals and plants[24]. Notably, the genetic basis of compatibility is generally different between the vegetative and sexual stage but some *het* genes are known to have pleiotropic effects on the sexual cycle of a number of species[25,26]. Of the nine genetically identified *het* loci in *P. anserina*, six are known to have antagonistic pleiotropic effects on the sexual function through sterility or progeny inviability[12,27] (Supplementary Fig. 1). In this study, we used population genomic analyses, laboratory crosses, genetic manipulations, field observations and simulations to show that two *het* genes define reproductively isolated groups in *P. anserina* and propose scenarios of how this arises.

## Results

**Low genetic diversity and balancing selection in *P. anserina*.** We started by sequencing the haploid genomes of 106 *P. anserina* strains, spanning 25 years of sampling around Wageningen, the Netherlands (from 1991 to 2016; Supplementary Table 1). Whole genome sequence data showed that the *P. anserina* samples are remarkably similar, with an average pairwise nucleotide diversity ($\pi$) of 0.000492 (Fig. 1a and Supplementary Fig. 2). As *P. anserina* does not produce asexual propagules, we expect clonality to be low but the extremely low genetic diversity could also be the result of very high selfing rates and demographic processes like bottlenecks. As an indication of the degree of outcrossing, we calculated the linkage disequilibrium (LD) decay for the Wageningen collection (Supplementary Fig. 3). We found that LD, as measured by the $r^2$ statistic, reaches 0.2 at distances <3.5 kilobases (kb) for most chromosomes, except for chromosome 4 that has a much slower decay ($r^2 < 0.2$ after 12.2 kb; Methods). These values are intermediate between those typical for outcrossers and for extreme selfers in fungi[28]. Additionally, we sampled 68 new genetically

[1]Systematic Biology, Department of Organismal Biology, Uppsala University, Uppsala, Sweden. [2]Department of Zoology, Stockholm University, Stockholm, Sweden. [3]Institut de Biochimie et de Génétique Cellulaires, UMR 5095, CNRS, Université de Bordeaux, Bordeaux, France. [4]Laboratory of Genetics, Wageningen University & Research, Wageningen, the Netherlands. [5]Department of Biology, Norwegian University of Science and Technology, Trondheim, Norway. [6]Plant Ecology and Evolution, Department of Ecology and Genetics, Uppsala University, Uppsala, Sweden. ✉e-mail: lorena.ament@zoologi.su.se; hanna.johannesson@ebc.uu.se

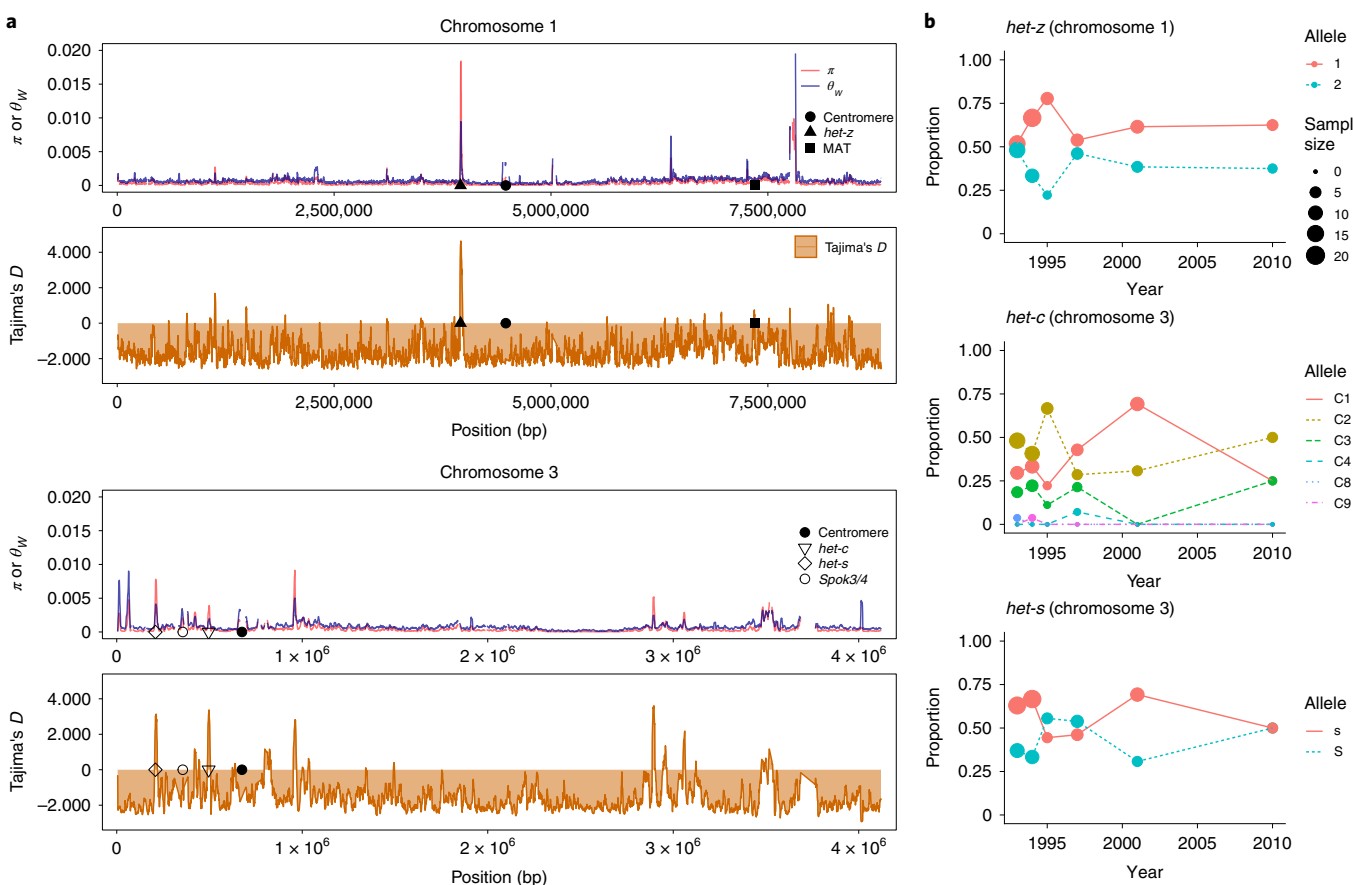

**Fig. 1 | Genetic diversity and balancing selection in the Wageningen collection of *P. anserina*. a**, Sliding window analysis (10 kb long with steps of 1 kb) of representative chromosomes 1 (top) and 3 (bottom) with values of genetic diversity (measured as either the pairwise nucleotide diversity $\pi$ in red or as Watterson's theta $\theta_W$ in blue) and the Tajima's *D* statistic (orange). The location of relevant loci is marked when present: *het* genes, the centromere, the mating type locus (MAT) and the location of meiotic drivers of the *Spok* family. Loci with highly divergent alleles (for example, MAT) and repetitive regions (for example, WD40 repeats of HNWD genes) were filtered out by the variant-calling pipeline but linked variants can still show signals of balancing selection. Note that *Spok3* and/or *Spok4* can be found at different locations in the genome depending on the strain. **b**, Changes of allele frequencies through time for three known *het* loci (data shown from years with more than five samples).

different strains in 2017 around Wageningen (Supplementary Table 2; Methods) and investigated the occurrence of spore killing, a phenotypic expression of meiotic drive in fungi. Previous studies have shown that natural populations of *P. anserina* harbour a number of meiotic drivers (selfish genetic elements that cause segregation distortion) belonging to the *Spok* gene family[29,30]. Mating between an individual with a meiotic driver and an individual without it results in the abortion of the spores that did not inherit the meiotic driver. Hence, observing spore killing is a direct indication of an outcrossing event[29]. We encountered one case of spore killing in our sample despite the low frequency of active spore killers in this population (<20%; Supplementary Table 1)[29]. Taken together, we conclude that, while showing indications of a high selfing rate, *P. anserina* outcrosses at detectable levels in nature.

In contrast with the observed general low genetic diversity, previous studies have found a great variety of vegetative incompatibility groups (defined by the *het* genes) within the *P. anserina* Wageningen collection[31,32]. Under the expectation that the *het* genes evolve under negative frequency-dependent selection[33–35], we calculated the Tajima's *D* statistic in sliding windows along the genome (Fig. 1a and Supplementary Fig. 2). As expected, the windows containing most *het* genes exhibit high genetic diversity and locate directly under high positive Tajima's *D* peaks (Fig. 1a and Supplementary Fig. 2), an indication of balancing selection. The *het* genes that do not

colocalize with positive Tajima's *D* values belong to the HNWD gene family (for example, *het-d* and *het-r* in chromosome 2; Supplementary Fig. 2), which are characterized by WD40 tandem repeats at the C terminus that define the allele specificity[36,37]. The repetitive end of the HNWD genes excluded them from our variant-calling pipeline, making genome scans inappropriate to assess signatures of selection for such genes. Nonetheless, other members of the HNWD family (for example, *het-e* and *hnwd3* in chromosomes 4 and 7, respectively; Supplementary Fig. 2) are flanked by sites with high Tajima's *D*, suggesting that at least some members of this gene family evolve under balancing selection.

To complement the signatures of historical demographic processes provided by the Tajima's *D* statistic, we took advantage of our temporal data and plotted the changes in allele frequencies through time for representative *het* genes in the Wageningen samples (Fig. 1b and Supplementary Fig. 4). As expected under balancing selection, the *het* genes show intermediate allele frequencies over the sampling time period. This pattern includes the *het-c* gene (Fig. 1b), which is the interacting partner of *het-d* and *het-e*[38], providing further support for the hypothesis that HNWD genes are also under balancing selection[31].

**Two reproductively isolated groups exist within *P. anserina*.**
Previously, van der Gaag[39] performed pairwise crossings of around

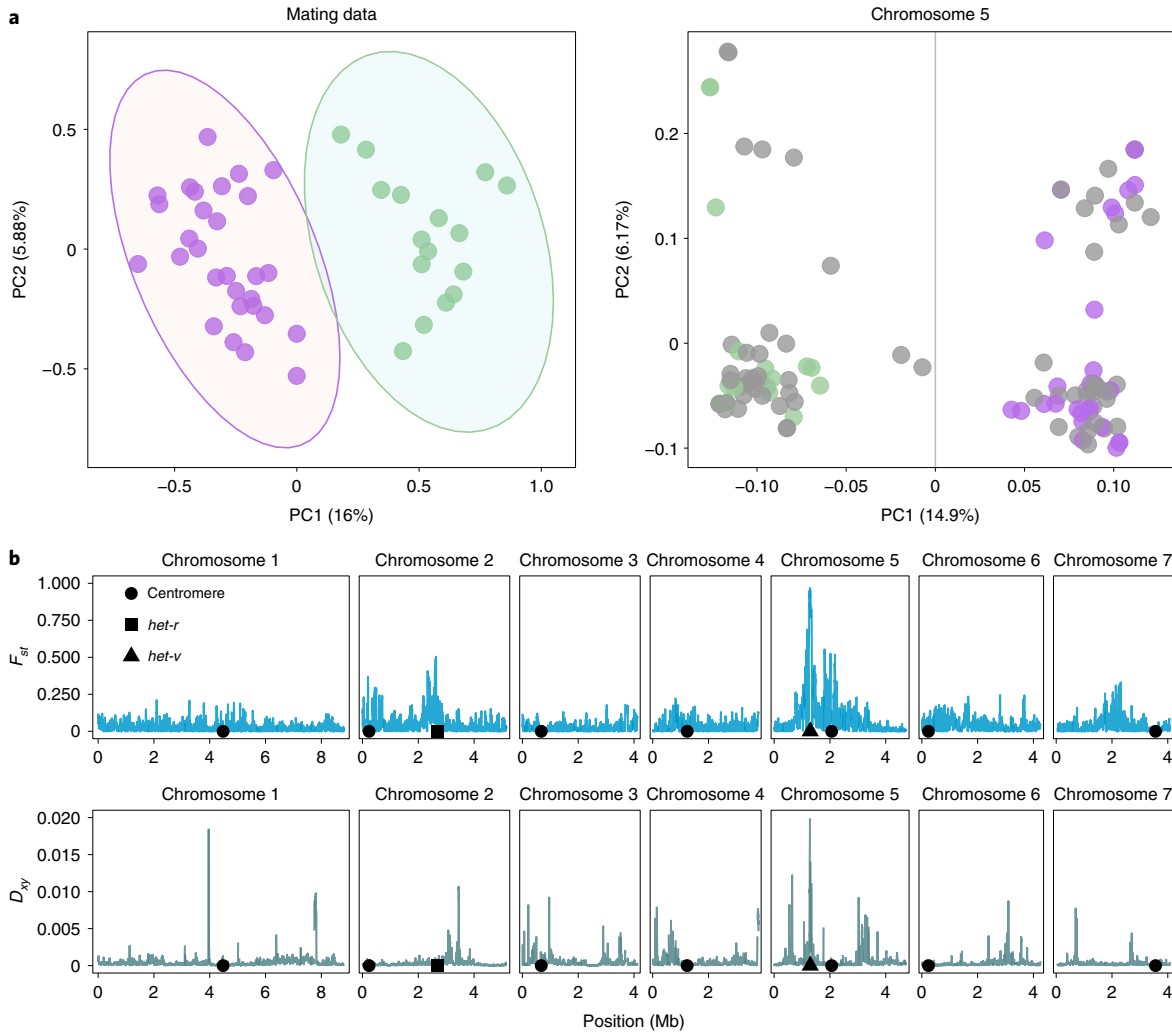

**Fig. 2 | Mating success correlates with genetic differentiation in chromosome 5. a**, Left, a PCoA of mating success data reveals two groups (indicated by lilac and green ovals); right, a PCA of the SNP data from chromosome 5 returns clusters of the same two groups (divided by the grey line). Samples with both mating and genomic data are coloured on the basis of the mating success clustering (lilac and green points). Samples without mating success data are in grey. **b**, Genetic differentiation between the two groups (as defined by the PC1 axis of the SNP data) shown as $F_{st}$ and $D_{XY}$ statistics estimated from sliding windows (10 kb long with steps of 1 kb).

half the Wageningen collection (plus the French reference strain S), recording various degrees of sexual and vegetative incompatibility. To gain insight into the population structure, we re-coded these observations into a distance matrix of mating success for 45 strains, for which we also have Illumina data (Supplementary Table 3). A principal coordinates analysis (PCoA) of the distance matrix revealed two groups, as identified by a partitioning around medioids (PAM) clustering method (Fig. 2a). Notably, the same two groups can be identified using a principal component analysis (PCA) of whole-genome single nucleotide polymorphism (SNP) data (Supplementary Fig. 5a). Moreover, analysis by chromosome reveals that the clustering signal is driven by variation on chromosome 5 (Fig. 2a) and to a lesser extent chromosome 2 (Supplementary Fig. 5b–d).

On the basis of the result of the clustering analysis, we divided all the samples into what we refer to hereafter as the two 'reproductively isolated' (RI) groups. For samples with no mating success data (59.8% of the total), we assumed RI group membership on the basis of the first principal component of chromosome 5 (Fig. 2a). We computed the population fixation index $F_{st}$ and the divergence statistic $D_{xy}$ between the RI groups in windows along each chromosome

(Fig. 2b, Supplementary Fig. 6 and Supplementary Fig. 7). While most chromosomes show virtually no differentiation, we found that a region near the centre of the left arm of chromosome 5 is strongly differentiated (nearly reaching the maximum $F_{st} = 1$ and having the highest $D_{xy}$ value). In accordance with the PCA, a region on chromosome 2 also shows intermediate to high levels of differentiation (on the basis of $F_{st}$ but not of $D_{xy}$). The area of differentiation in chromosome 2 co-localizes with the *het-r* gene, one of the *het* genes with pleiotropic effects on sexual reproduction (Supplementary Fig. 1). From classical genetic analyses, it is known that *het-r* works as an allorecognition gene through its interaction with the uncharacterized *het-v* locus[12], which is localized on chromosome 5 (ref. [40]). Hence, although other *het* genes cause sexual incompatibilities, this interacting pair is the most likely functional candidate behind the partition into two RI groups and the associated genomic divergence. We thus focused specifically on these two loci in further analyses.

Like other members of the HNWD family, the *het-r* gene has a WD40 repeat domain at its C terminus[37]. The number of tandem WD40 repeats and their sequence have been shown to define two alleles: *R* and *r* (ref. [36]). Allele *R* has 11 WD40 repeats in a specific order. In effect, allele *r* is a non-functional variant that can have any

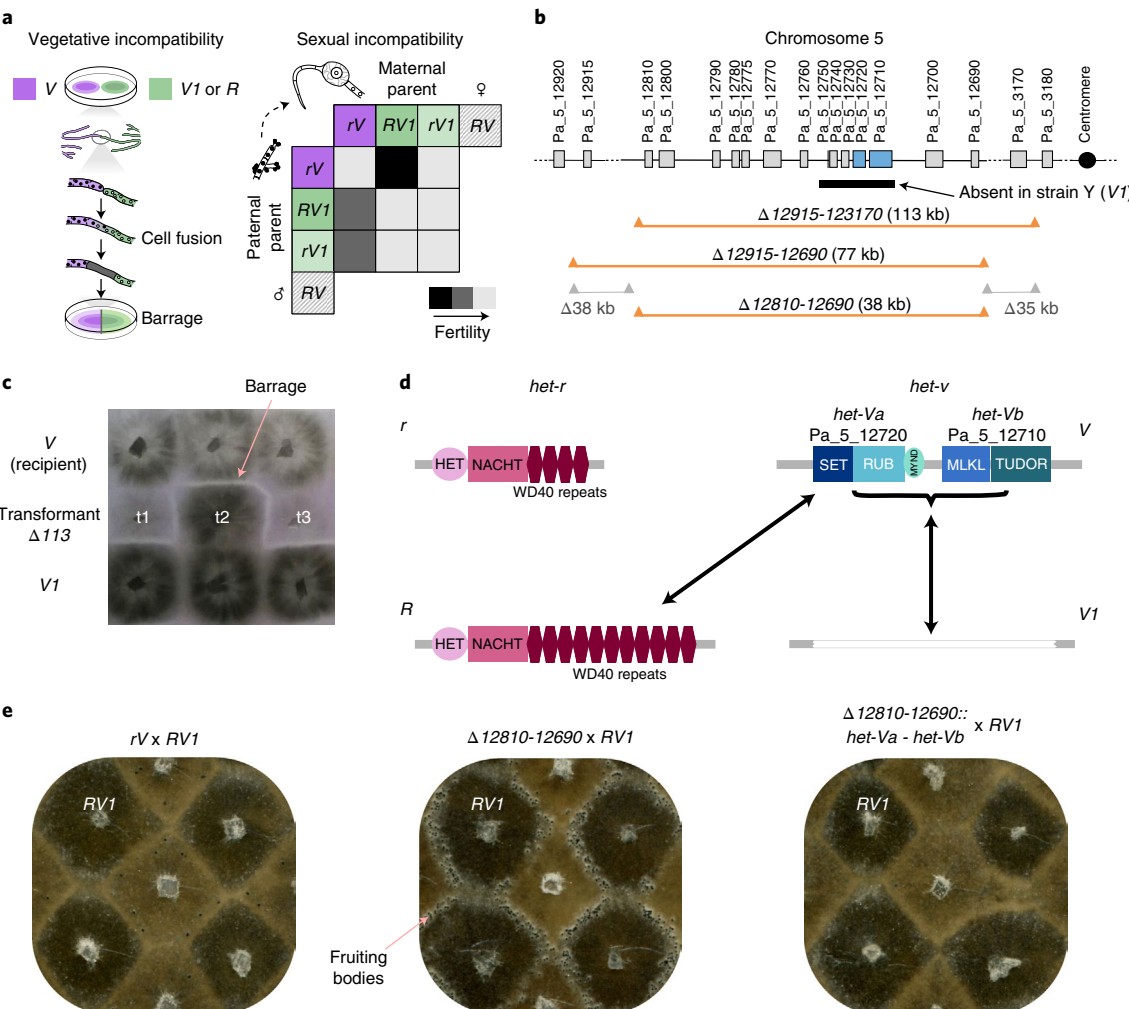

**Fig. 3 | Genetic and molecular dissection of the *het-v* locus. a**, When two individuals with incompatible alleles fuse during vegetative growth, an incompatibility reaction occurs producing a divisive line of dead cells known as barrage. During sex, when the female organ (protoperithecium) is fertilized by microconidia, different genotypes of the two *het* genes confer various degrees of infertility depending on the crossing partner. The *RV* genotype is lethal. **b**, The location of *het-v* was identified through nested deletions, marked in orange if they lost the barrage formation in confrontations with a *V1* strain (that is, became compatible) or in grey if they did not. The black bar represents a cluster of genes absent in the strain Y, which is of incompatibility type *V1*. **c**, Deletion *Δ113* (*Δ12915-123170*) from a *V* strain leads to the *V1* phenotype. A barrage test with a *V* (top row) and *V1* tester (bottom row) is given for three transformants obtained with the *Δ113* deletion cassette using an *rV* recipient strain. The central transformant t2 produces a barrage reaction to *V* but not to *V1* (that is, it acquired the *V1* phenotype). Transformants t1 and t3 retain the *V* phenotype and are presumably *V + V1* heterokaryons. **d**, Genetic structure of the *het-r* and *het-v* loci is represented by the domain architecture of the encoded proteins. Arrows represent incompatible interactions. **e**, To the left, a cross between *rV* (light colour) and *RV1* (dark colour) produces very few fruiting bodies. However, fertility is recovered by deleting the *het-v* locus and adjacent genes (*Δ12810-12690*) as shown in the cross in the middle. Reintroduction of the *het-Va* and *het-Vb* genes fully restores the sterile phenotype characteristic of *rV × RV1* in the cross to the right.

other number of repeats or a mutated version of the basic 11 repeats[36]. The *het-v* locus also has two alleles, *V* and *V1*, as defined by classical genetics[41]. The vegetative incompatibility reaction (or barrage) is triggered when individuals of different *het-v* alleles meet and fuse (Fig. 3a). Different *het-r* alleles do not trigger barrage formation on their own but there is a non-allelic interaction between the alleles *R* and *V* leading to a barrage. Crucially, the allelic and non-allelic vegetative interactions are mirrored by sexual sterility with severity depending on the genotype of the parents[42,43]. Moreover, individuals of the genotype *RV* are self-incompatible, making that combination lethal upon germination[41,44] (Fig. 3a). While the *het-r/v* incompatibility had been previously linked to sexual dysfunction, there had been no indication of this leading to two RI groups as observed here. To confirm that *het-r/v* are indeed responsible for causing RI in *P. anserina*, we molecularly characterized the *het-v* locus.

**Locus *het-v* encodes two genes that cause incompatibilities.** As the *het-v* locus was genetically mapped to the left arm of chromosome 5, we introduced genetic markers in that region and analysed their linkage to the vegetative incompatibility phenotype of *het-v* (Supplementary Fig. 8; Methods). This led to the identification of a 113 kb long candidate region (Fig. 3b). Deletion of this region in an *rV* strain led to loss of the barrage reaction to an otherwise isogenic *rV1* strain, as expected if *het-v* locates there. Notably, while losing the *V* phenotype, these deletion strains simultaneously acquired the *V1* phenotype (rather than showing a neutral incompatibility phenotype) (Fig. 3c). In other words, deleting the region with the *V* allele resulted in vegetative incompatibility to *V* and compatibility to *V1*.

Using additional nested deletions in the *rV* strain (Fig. 3b and Supplementary Fig. 8) we narrowed down the candidate region

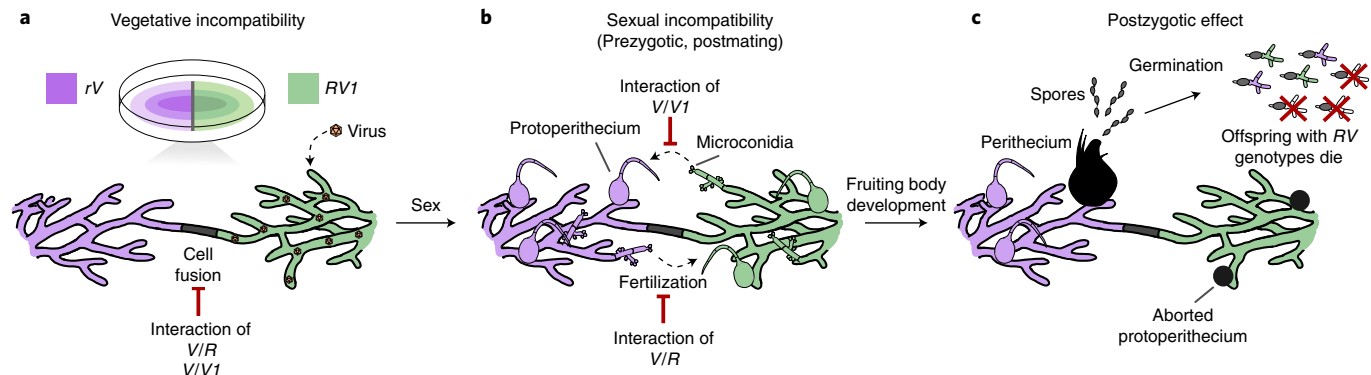

**Fig. 4 | The effects of the *het-r/v* system on vegetative incompatibility and reproductive isolation. a**, Vegetative cell fusion between members of the two RI groups is prevented by the allelic interaction of the *het-v* alleles (*V/V1*) and the non-allelic interaction between *het-v* and *het-r* (*V/R*), which deters the transfer of viruses and other deleterious cytoplasmatic elements. **b**, During sex, the *het-r/v* system remains active, resulting in failed fertilization. **c**, In most cases the zygote formation does not occur and female organs (protoperithecia) do not develop into perithecia. The few successful perithecia produce spores but a fraction of these spores are self-incompatible and die 15 h after germination[12,41–44]. While female *R* organs are irreversibly damaged after abortive fertilization with *V* microconidia, female *V* organs are still viable after fertilization with *R* (or *V1*) microconidia[41,43].

to a 38 kb area (between the genes Pa_5_12810 and Pa_5_12690, henceforth *Δ12810-12690*; ref. [45] gives gene codes). Inspection of the sequenced genome of the wild-type strain Y, which is of the *V1* incompatibility type, revealed that a ~12 kb region spanning five genes (Pa_5_12750 to Pa_5_12710) is replaced by a cluster of transposable elements when compared with the reference genome of the S strain[45], which is of the *V* type. Consistent with the deletion strain experiment, these five genes are contained within the *Δ12810-12690* area. The *V1* phenotype of the deletion strains, in turn, indicates that the cluster of transposable elements at the wild-type *V1* locus is not required for the *V/V1* incompatibility.

To determine which of the five open reading frames (ORFs) within the region absent in the strain Y are responsible for the *V* phenotype, we turned to a complementation cloning approach using the *rV Δ12810-12690* strain (which displays the *V1* phenotype) as a recipient. Our results showed that insertion of the genes Pa_5_12710 and Pa_5_12720 confers the *V* phenotype on the recipient strain and that these two genes thus correspond to *het-v*. Furthermore, both genes together are required for allelic incompatibility to *V1*, while Pa_5_12720 alone determines non-allelic incompatibility to *R* (Fig. 3d and Supplementary Table 4). A strain expressing Pa_5_12720 (but not Pa_5_12710) becomes compatible with both *V* and *V1*. Thus, we propose to name Pa_5_12720 and Pa_5_12710 as *het-Va* and *het-Vb*, respectively (Fig. 3d).

Since *het-Va* encodes a predicted lysine or histidine methyltransferase (Fig. 3d and Supplementary Fig. 9), we hypothesized that this activity was required for the incompatibility function. Two different point mutants of the catalytic tyrosine residue were obtained (Y233A and Y233F; Supplementary Fig. 9). When introduced into a recipient with the *rV1* phenotype (*V Δ12810-12690*), both mutants restored vegetative incompatibility to *R* (Supplementary Table 4). In contrast, co-transformation of *het-Vb* with *het-Va* Y233A or Y233F failed to restore the barrage reaction to *V1*. We conclude that the methyltransferase activity of HET-Va is required for the allelic *V1/V* vegetative incompatibility but dispensable for non-allelic *R/V* vegetative incompatibility (Fig. 3d).

To confirm that the vegetative and sexual effects associated with *het-v* have the same mechanistic basis, we verified that the *Δ12810-12690* deletion of the region encompassing *het-Va* and *het-Vb*, which converts *V* to the *V1* vegetative phenotype, also restores fertility in crosses to an *RV1* strain (Fig. 3e and Supplementary Fig. 10). We found that a *Δ12810-12690 × RV1* cross shows normal fertility. Moreover, when *het-Va* and *het-Vb* are re-introduced by

transformation into *Δ12810-12690* strains, the crosses to an *RV1* strain show sterility. In addition, we found concordance between the sexual and vegetative phenotypes when the *het-Va* point mutants were used for transformation (Supplementary Fig. 11 and Supplementary Table 5). An *rV1* recipient transformed with *het-Va* Y233F (or Y233A) together with *het-Vb* showed normal (female) fertility to *rV1*, while transformants expressing *het-Va* Y233F remained sterile (as males) in crosses to *RV1*. Thus, as for the vegetative incompatibility, the methyltransferase activity is dispensable for *R/V* but required for *V/V1* sexual incompatibility. However, a discrepancy between the vegetative and sexual effects was noted for the *het-Va* Y233A mutation, which suppressed *R/V* sexual but not vegetative incompatibility. It is possible that this less conservative substitution partially destabilizes the *het-Va* product and that decreased protein amounts are sufficient to trigger vegetative but not sexual incompatibility. Accordingly, it has been observed in other *P. anserina het* systems that higher protein amounts are required for sexual than for vegetative incompatibility[46]. Thus, we conclude from these experiments that *het-Va* and *het-Vb* are responsible both for the vegetative and sexual phenotype of *het-v* and that sexual incompatibility is not the result of linked variants.

In essence, because the cell death reaction is not turned off during sexual reproduction, the *het-r/v* system acts as a reproductive barrier (Fig. 4; also Supplementary Fig. 12). Asymmetry in fertilization probably arises from the fact that the *V* allele is a diffusible cytoplasmic factor, while the HET-R protein is not[41,43]. Potentially, the HET-R protein is found in very low amounts or does not diffuse from the male gamete (microconidium) to the female organ (protoperithecium). As a result, the *R* protoperithecium is damaged by the *V* products but not vice versa (Fig. 4c).

In combination with previous knowledge on *het-r*, we can now derive a mechanistic model for *het-r/v* allorecognition function. It is known that HET-R has a tripartite domain organization typical of nucleotide-binding oligomerization domain (NOD)-like receptors or NLRs[23,37]. Thus, the *het-r/v* interaction can be hypothesized to function analogously to other incompatibility systems involving NLRs such as *het-c/e/d*, *het-z* and the *het-s/nwd2* systems[31,47,48]. In such a model, HET-R would bind the HET-Va protein via the variable C-terminal domain of WD40 repeats, which would trigger oligomerization of its NACHT domain and downstream activation of the HET cell death-inducing domain. This model accounts for the fact that WD40 repeat loss in HET-R leads to the inactive

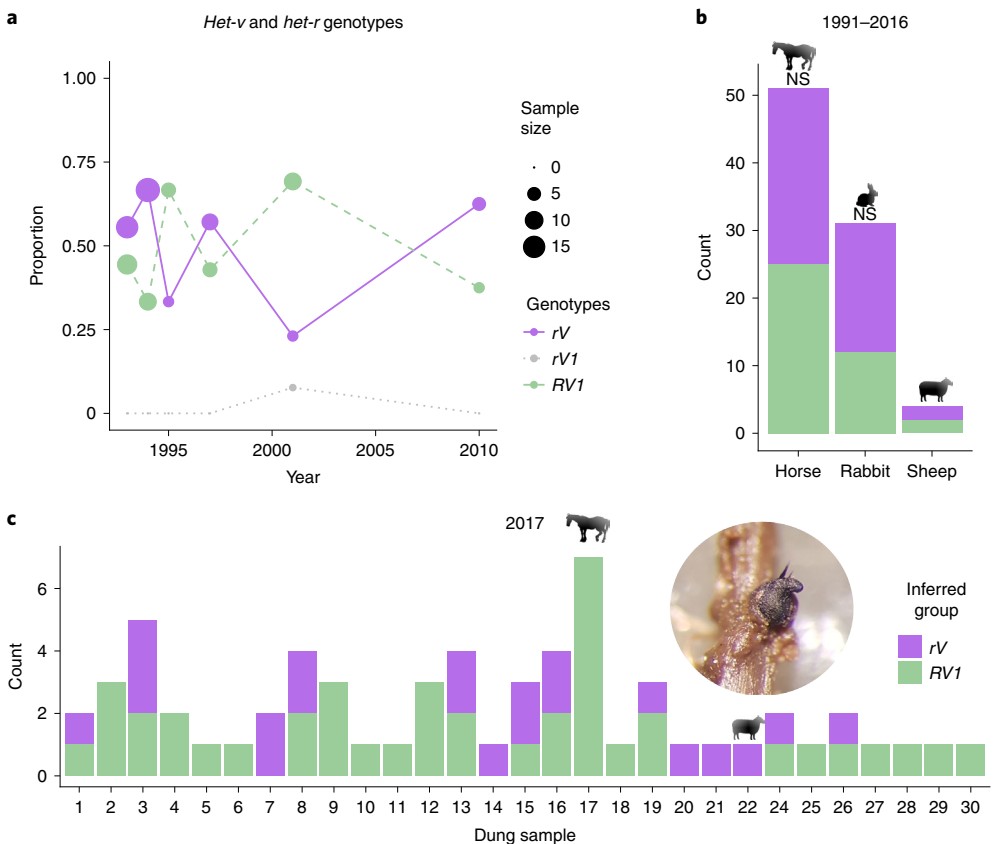

**Fig. 5 | The RI groups maintain isolation despite coexistence. a**, Genotypes of *het-v* and *het-r* in the Wageningen population through time. The *rV1* genotype is very rare, confirming that the RI groups (*RV1* and *rV*) are not mixing freely. *RV* is lethal. **b**, There is no significant difference (NS) in abundance between RI groups based on the source herbivore (Pearson's $X^2_{1,82}=0.46365$, $P=0.4959$; genotyped samples from 1991 to 2016 with substrate data, $n=82$). Sheep dung was not tested due to low sample sizes. **c**, The strains collected in 2017 from horse dung (plus one sample from sheep) show that the RI groups can be found in the same dung piece ($n=63$). Insert, fruiting body of *P. anserina* in the wild (photo by S.L.A.V.).

*r* phenotype[36] and that *R/V* incompatibility is independent of the predicted methyltransferase activity of HET-Va.

The genetics of the *V/V1* incompatibility, by contrast, represents a rather puzzling situation from a mechanistic point of view. It is unclear how the *het-Va* and *het-Vb* genes (that is, the *V* allele) are able to sense the absence of their own products (*V1*) during the vegetative (and sexual) incompatibility reaction. HET-Va is a predicted methyltransferase, while HET-Vb displays a TUDOR domain occurring in so-called reader proteins, which recognize the methylation marks deposited by methyltransferases[49], as well as a MLKL/HeLo membrane-targeting cell death-inducing domain[50]. The domains involved suggest that incompatibility could be brought about by a combination of methyl mark deposition (by HET-Va) and reading by the HET-Vb TUDOR domain and subsequent activation of the MLKL/HeLo domain ensuring cell death execution (Supplementary Fig. 13), by analogy to the mechanism of cell death execution in mammalian necroptosis or *het-s* incompatibility[50].

At any rate, regardless of the mechanism of *V/V1* incompatibility, the implication of the HET-R NLR stands out because NLRs are key components of the innate immune system of plants and animals[23,51,52]. Immunity genes, like the *het* genes, are expected to evolve under negative frequency-dependent selection, although driven by pathogen pressure instead of vegetative fusion. Bateson–Dobzhansky–Muller incompatibilities[53] can arise between populations adapted to different pathogen pools, which in plants can result in the autoimmune response known as hybrid necrosis[54]. Crucially, characterized hybrid necrosis genes happen to be NLRs themselves in, for example, *Arabidopsis thaliana*[55], tomato[56] and cotton[57]. Thus,

genes of similar molecular characteristics and selective pressures driven by non-self recognition can contribute to reproductive isolation in vastly different taxa.

**The *het-r* and *het-v* loci induce reproductive isolation in the wild.** Once the molecular basis of vegetative and sexual incompatibility was identified, we set out to further characterize the Wageningen population with regard to the two RI groups. Although the effects of the *het-r/v* interaction on sexual incompatibility are significant, they are not absolute. Hybrids can be produced in the laboratory (Fig. 3e, left) and the *RV* reaction is thermosensitive, losing effect at 32 °C (ref. [58]). To evaluate if the RI group effects we observed in the laboratory crosses hold in wild populations, we analysed the population genomic dataset to determine their *het-r/v* alleles. We reason that if, despite sexual incompatibilities, the two RI groups are mixing in nature, we should encounter the three possible viable genotypes between *het-v* and *het-r* (that is, *rV*, *RV1* and the recombinant *rV1*; Fig. 3a) in the wild, simply because of independent chromosomal segregation. We confirmed that the RI groups (as in PC1 in Fig. 2a) are perfectly defined by the *het-v* allelic identity. Indeed, the maximum values of $F_{st}$ occur at the *het-v* locus (Fig. 2b and Supplementary Fig. 6). Since the members of the HNWD family cannot be assembled with Illumina data, we used a PCR-based method of ref. [59] to genotype the *het-r* gene in the Wageningen collection (Supplementary Tables 1 and 6). As expected, under strong reproductive isolation, we found a nearly perfect association of *r* with *V* and *R* with *V1* throughout the sampled years (Fig. 5a), with an LD estimate of $r^2=0.926$ between the two loci. By contrast, *het-v*

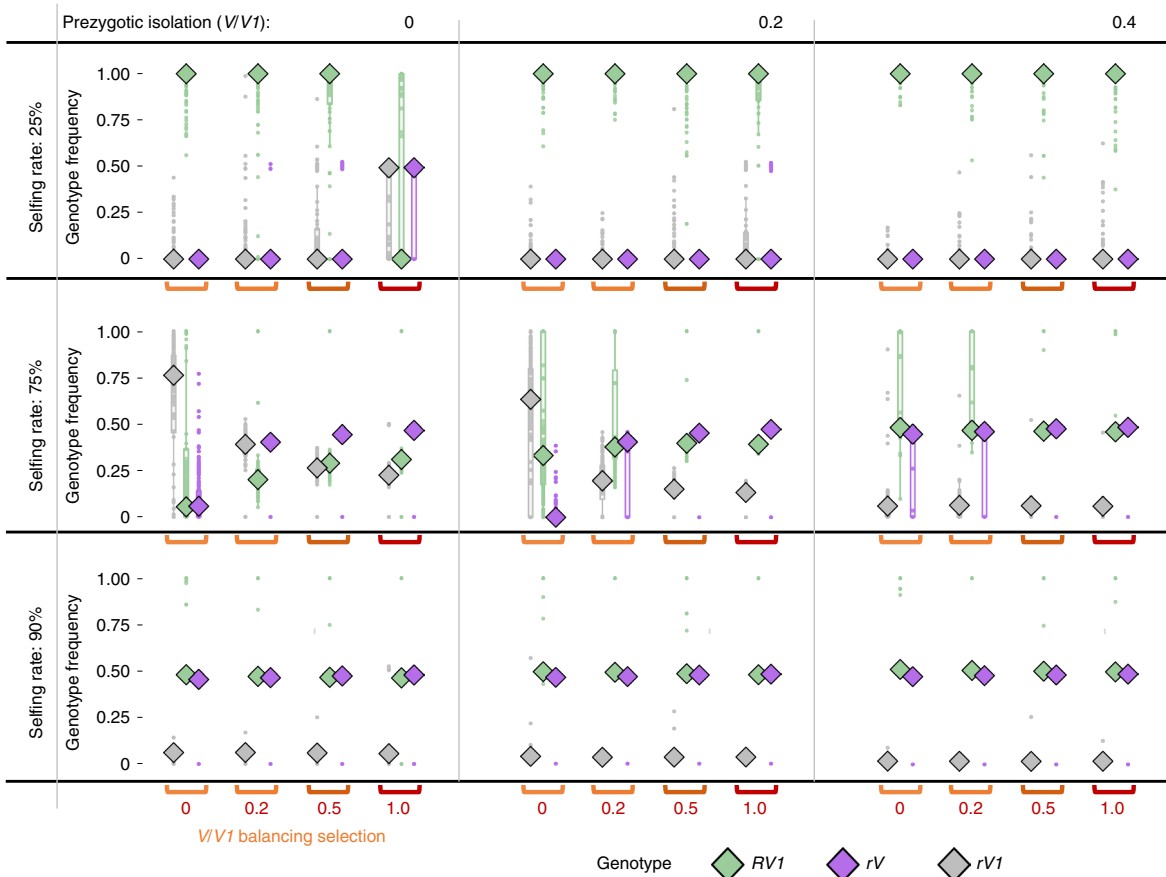

**Fig. 6 | Summary of individual-based simulations of *rV* invading an *RV1* population with intensity of *V/R* balancing selection fixed to 0.5.** For each parameter combination, the distribution of genotype frequencies of 100 replicated simulations is given. Each simulation is represented by a dot and their distribution by a boxplot, which shows the median as a diamond, the 25th to 75th percentiles as the box bounds and 1.5× interquartile ranges as whiskers. If dots and boxplots are not visible, the distribution is concentrated behind the diamond of the median.

has no association with other *het* genes, such as *het-z* in chromosome 1 ($r^2 = 0.004$) or *het-q* in chromosome 7 ($r^2 = 0.009$), showing that the LD is not genome-wide (Supplementary Fig. 14a). We further found a modest association of sites around (linked to) *het-v* and the area of chromosome 2 surrounding *het-r* (Supplementary Fig. 14b), further supporting the association between the *het-r/v* genotypes.

One may hypothesize that the lack of mixing between RI groups is strengthened by differences in ecology, such as substrate or temporal isolation. Like other coprophilous fungi, *P. anserina* has a short life cycle (completed in around 11 d under laboratory conditions), where the sexual spores are ejected into the surrounding vegetation and subsequently ingested by herbivores. The spores go through the animal's digestive tract and germinate, grow and reproduce within the dung (Supplementary Fig. 15). From the genotyped strains with substrate information available, we saw no significant differences in the abundance of each RI group (for horse and rabbit dung: Pearson's $X^2_{1, 82} = 0.46365$, $P = 0.4959$; Fig. 5b). To assess the possibility of temporal isolation, we phenotypically assigned strains collected in 2017 to each RI group by confronting them with tester strains of known genotype. Remarkably, the RI phenotyping matched perfectly the expected *het-v* allele in a subset of samples evaluated with PCR (Supplementary Table 2). We simultaneously confronted strains sampled within the same dung piece to each other and used barrage formation to estimate the number of distinct individuals present. We found that more than one individual can be found in the same dung piece and that such individuals often belong

to different RI groups (Fig. 5c). Taken together, this demonstrates that the RI groups do not lack opportunities for mixing in the wild.

**Simulations recapitulate the formation of two RI groups.** To investigate the conditions under which a split of a population by means of pleiotropic genes under balancing selection can occur, we designed an individual-based simulation using SLiM[60] on the basis of the *Podospora* life cycle and our characterization of the *het-r/v* interaction (Supplementary Methods and Supplementary Fig. 16). Since we do not know the ancestral state of either *het* gene, we used as starting point populations of a single genotype, either *rV* or *RV1*, and then a small fraction of individuals of opposite genotype were introduced. Regardless of the starting genetic composition, we found that balancing selection and some degree of selfing is required for the introduced genotype to invade the population (Fig. 6 and Supplementary Figs. 17 and 18). Selfing is needed because the invader allele immediately suffers the cost imposed by the lethal *RV* combination produced during outcrossing, which will prevent it from increasing in frequency. In the presence of selfing, the cost can be avoided and balancing selection then provides benefits to the invader alleles through rare-allele advantage, eventually driving them to intermediate frequencies. Importantly, the population is composed of the two expected genotypes (*rV* and *RV1*, to the exclusion of *rV1*) under several scenarios, most of them dependent on the strength of balancing selection acting on the non-allelic (*R/V*) system and secondarily on the allelic one (*V/V1*) (Fig. 6 and Supplementary Figs. 17 and 18). In addition, high levels of

selfing, prezygotic isolation or both, can lower the frequencies of the *rV1* genotype, completing isolation. Notably, a theoretical study on hybrid necrosis genes in plants also found a role of negative frequency dependence on the build-up of reproductive barriers[61], with the difference that the isolating barrier in question is exclusively postzygotic, instead of prezygotic and postzygotic as in the case of *P. anserina*. Their model also included selfing but the authors did not explore the effect of this parameter on their results. Overall, we conclude from the simulations that RI groups can evolve from pleiotropic allorecognition loci in organisms that undergo selfing (or inbreeding) at high rates during the invasion of one of the two incompatible genotypes. Analogous to the scenario where pleiotropic effects of the MHC drive speciation through assortative mating[11], in *Podospora* selfing acts as the non-random mating process that allows reproductive isolation. Moreover, sympatry is a necessary condition for the evolution of reproductive isolation in our model, since vegetative interactions between individuals of the two groups are required for balancing selection to operate.

Of note is the fact that, out of all the *het* systems of *P. anserina* that have effects on the sexual function, only *het-r/v* leads to detectable effects on population structure and compatibility type differentiation. Both the *het-c/e/d* and the *het-z* systems lead to sterility in specific combinations of *het*-genotypes in the male and female parent (Supplementary Fig. 1)[12]. Likewise, the unlinked loci *het-c/e/d* can produce 'self-incompatible' progeny that die after germination, just like *het-r/v*[12,42]. However, our simulations show that, in the absence of balancing selection acting on an allelic system and the associated prezygotic incompatibilities (roughly equivalent to a simplified *het-c/d* or *het-c/e* interaction, for example), the two mating groups can only evolve under extremely high selfing rates. Probably, the fact that sterility is (nearly) symmetrical in the *het-r/v* system, but not in the other ones, makes it particularly prone to split the population.

Admittedly, our simulations do not fully capture the complexity of the multiallelic *het-c/e/d* system, where sexual incompatibility can be additive. For instance, a *C1E1 × C2E2* cross is nearly totally sterile, as their *het-c* and *het-e* alleles are incompatible. Yet, there are numerous wild-isolates that show neutral *het-d* and *het-e* alleles[12]. Presumably, the strains with neutral *het-d* and *het-e* alleles can bridge the gene flow between incompatible isolates, limiting their effects on reproductive isolation. Interestingly, the *het-c* alleles that can interact with more *het-e/d* alleles and which would be more effective at inducing reproductive isolation, have a low frequency in the Wageningen population (alleles *C4*, *C8* and *C9* in Fig. 1), suggesting that they are too costly relative to their adaptive value[31].

**The *het-v* locus is physically close to a meiotic drive gene.** In addition to the factors discussed above, meiotic drive could further influence the maintenance of allorecognition genes with pleiotropic effects. The advantage provided by segregation distortion can counteract selection against deleterious alleles, leading to stable polymorphism[62]. We genotyped all samples for the presence of members of the *Spok* family (Supplementary Table 1), confirming previous indications[29] that the *Spok2* gene is at a high population frequency (86.7%). The location of *Spok2* is relatively close to *het-v* and falls within the area of significant differentiation between RI groups (Supplementary Figs. 6 and 14). If the presence of *Spok2* predates the evolution of *het-r/v*, the segregation distortion advantage of *Spok2* could have aided the establishment of the incompatibility system despite the cost of sexual incompatibility. Alternatively, *Spok2* could have invaded *P. anserina* after the evolution of *het-r/v*. As spore killers generally only experience the fitness boost from meiotic drive after reaching some minimum population frequency[63], *Spok2* may have benefited from hitchhiking along with a locus under balancing selection at early stages of population invasion. Recombination could have later decoupled *Spok2* from a given *het-v* allele,

allowing it to invade both RI groups and approach fixation. The invasion of *Spok2* could have then eroded divergence between RI groups, as observed in meiotic drive systems of *Drosophila* species[64]. Data from different populations of *Podospora* are required to clarify this link but, regardless, the proximity of *Spok2* and *het-v* hint to a previously unappreciated way meiotic drive could contribute to speciation.

**The *het-r* and *het-v* loci are present in related species.** Finally, in an attempt to discern the evolutionary history of *het-r* and *het-v* in *P. anserina*, we examined the genomes of other members from the *P. anserina* species complex[65]. We found that orthologues of *het-r* are present in all seven species of the complex, although their allelic identity is unknown (Methods). In the case of *het-v*, we found that *het-Va* and *het-Vb* are present in at least some species but there is also evidence of introgression of *het-v* between *P. anserina* and *P. pauciseta* (Supplementary Fig. 19). These observations suggest a dynamic evolution of the *het-v* locus throughout the divergence of this species complex but without population sampling from the other *Podospora* species we cannot draw concrete conclusions.

Looking at other genera of the order Sordariales[66], we also found that *het-Va* and *het-Vb* show conserved synteny in *Sordaria macrospora* and a number of *Neurospora* species (Supplementary Table 6). As large genomic datasets of the *Neurospora* species are available, we investigated the distribution of the *het-v* alleles across the genus. We found that only *Neurospora perkinsii*, *N. intermedia* and some lineages of *N. tetrasperma* have full copies of both *het-Va* and *het-Vb*. In *N. crassa* and *N. sitophila*, the gene *het-Vb* seems pseudogenized, while in *N. metzenbergii* and *N. tetrasperma* lineages L4, L7 and L8 (ref. [67]) both genes are completely absent. Notably, *N. intermedia* and some lineages of *N. tetrasperma* are polymorphic for pseudogenized and full copies of *het-Vb*, raising the question of whether *het-v* also works as an allorecognition gene in these species. Furthermore, different taxa have their own premature stop codons, suggesting independent loss events. Thus, although we lack mechanistic evidence for the function of these genes in *Neurospora*, the observed patterns suggest that *het-v* might be old but involved in rapid presence/absence turnover in other taxa. By contrast, we found that *het-r* gene does not appear to be present in *Neurospora*.

Importantly, the *het-r/v* interaction itself cannot be ancestral to the *P. anserina* species complex, as the *het-r/v* incompatibility leads to speciation and thus to the loss of polymorphism (that is, each genotype is 'fixed' in each RI group). Accordingly, the extremely low diversity and the lack of differentiation along the genome (based on both $F_{st}$ and $D_{xy}$), other than around *het-v* and *het-r*, suggest that the RI group isolation is extremely recent. In addition, as it is known that there is a thermosensitive component to the *R/V* interaction[44], it remains of interest to study the phylogeography of this incompatibility system.

## Conclusion

Dissecting the genetic basis of reproductive barriers is a major challenge in speciation research[68,69]. In *P. anserina* we found that the *het-r/v* system not only contributes to speciation but it directly defines RI groups by conflating vegetative recognition with sexual compatibility. If current conditions are maintained, the *P. anserina* RI groups may accumulate further genetic incompatibilities and evolve ecological differences, eventually completing the speciation process. While the typical number and effect sizes of speciation genes (sensu ref. [70]) at the onset of speciation remains an open question, the case of this fungus suggests that speciation can happen through few loci of big effects, maintained by balancing selection forces and potentially high selfing or inbreeding rates. Moreover, the universality of the non-self recognition genetic systems implies that this type of reproductive isolation might occur in a wide variety of taxonomic groups.

## Methods

We used the packages ggplot2 v.3.0.0 (ref. [71]), cowplot v.1.0.0 (ref. [72]), gridExtra v.2.3 (ref. [73]), hexbin v.1.27.3 (ref. [74]), rstatix v.0.7.0 (ref. [75]) and ggpubr v.0.4.0 (ref. [76]) for data visualization and analysis. Animal silhouette figures were taken from http://www.printableparadise.com/printable-animal-silhouettes.html. Most bioinformatics analyses were performed using the workflow manager Snakemake v.5.4.4 (ref. [77]).

**Fungal material.** Including eight previously sequenced strains[29], our genomic dataset consisted of 106 *P. anserina* strains sampled in Wageningen, the Netherlands, and kept at the Laboratory of Genetics of the Wageningen University and Research[32,78–80]. These strains are identified with 'Wa' followed by a unique strain ID number (Supplementary Table 1). Most of the strains were obtained between 1991 and 2010 by isolating a single spore from a fruiting body (perithecium) in herbivore dung. Each spore was grown and selfed one or two times to obtain sexual spores that were stored at −80 °C. We further expanded the collection by sampling during 2016 and 2017 (see below) but the strains from 2017 were not sequenced. In addition, we included whole genome sequences of the *P. anserina* French strains S, Y and Z sampled in 1937[81,82], the strains CBS433.50 (Canada, 1944), CBS455.64m (Switzerland, 1964), $T_G$ (probably from France; see ref. [29]), as well as the *Podospora pauciseta* strain CBS 333.63 (Argentina, 1963). All strains that do not belong to the Wageningen collection have been selfed in the laboratory an unknown, but presumably large, number of times. For strains collected before 2016, we grew a new dikaryotic strain from the frozen stock, selfed it and isolated self-sterile monokaryotic (haploid) spores for sequencing and crosses (Supplementary Fig. 15). From 2016 and 2017, the monokaryons were isolated directly from the first laboratory selfing event. Notice that while the artificial selfing prevents an accurate assessment of heterozygosity, it should not alter the overall diversity in the collection. The mating type of the monokaryon isolates was determined by crossing with tester strains and annotated with a '+' or '−' sign.

**Sampling of new *P. anserina* strains.** We collected dung of herbivores (horse, sheep, cow and rabbit) around Wageningen during the months of September and October of 2016 and 2017. Small portions of dung were placed on top of wet filter paper on 9 cm plastic Petri dishes and incubated at room temperature for 2 weeks. Big pieces of dung (mostly from horse) were further fragmented into three to five replicate plates. Humidity was kept by adding a few drops of water (2016) or by keeping the filter paper on top of water agar (2017). After ~10 d of incubation, we observed the emergence of fruiting perithecia resembling *P. anserina* in 2 and 30 of the dung samples for 2016 and 2017, respectively (replicate plates were counted as one). Individual perithecia were recovered from the dung and placed on a water agar plate covered by an NC45 membrane filter of Schleicher & Schuell. Using a sterilized needle, we opened the fruiting body, confirmed four-spored asci and proceeded to recover the four spores of a single ascus whenever possible. The single spores were germinated for up to 2 weeks on a plate of $PASM_2$ media with 5 g l$^{-1}$ ammonium acetate added[29]. At 2 d postgermination, mycelia was transferred to $PASM_{0.2}$ plates, incubated for 3–5 d and stored. Only one spore per perithecium was used in all tests (all viable siblings were kept as a backup in the Wageningen collection). In total we sampled 4 (2016) and 68 (2017) strains, from which 5 are probably members of the sibling species *P. comata* (Supplementary Table 2). No strains were recovered from cow dung. During sampling in 2017, we observed spore killing in the asci of one fruiting body. We defined the identity of this spore killer in strain Wa224 as *Psk-7* (containing *Spok2*, *Spok3* and *Spok4*) on the basis of previously described methodology[29,80].

**Culture and crossing conditions.** Standard methods for growth and manipulation of *P. anserina* used here are described in the *Podospora* Genome project homepage (http://podospora.i2bc.paris-saclay.fr/) and in ref. [29]. Briefly, strains were grown on Petri dishes with either DO or Henks Perfect barrage (HPM) medium[29]. Crosses were done either by confronting monokaryotic strains growing together in the same plate or by spermatization. In the latter, mycelia of monokaryotic female parents are fertilized with a suspension of spermatia (microconidia) from a monokaryotic strain used as male parent. Likewise, vegetative compatibility was assessed through barrage formation upon confrontation. Barrage was assessed against a source of light. All cultures were kept at 27 °C with 70% humidity for a 12:12 light:dark cycle. Monokaryotic strains were stored on plates with $PASM_{0.2}$ media, a calorie-restricted $PASM_2$ recipe[83] with 0.2% glucose instead of 2% ($PASM_{0.2}$) to delay senescence of the isolates.

**RI group phenotypic assignment.** Wild strains of *P. anserina* were isolated from horse and sheep dung from Wageningen in 2017 (Supplementary Table 2). Multiple strains could be isolated from the same dung sample. To determine if these isolates represented unique strains or separate isolations of the same clone, all isolates from the same dung sample were confronted against each other under standard conditions on HPM medium. Formation of a barrage between strains was used to indicate that the isolates represented unique genets. If no barrage was formed, isolates were considered to represent the same strain. To confirm that the isolates represented strains of *P. anserina*, they were confronted against monokaryotic

tester strains Wa63 (*RV1*) and S (*rV*). New strains can be assigned to RI groups by evaluating the production of perithecia, whereby strains of the same RI group produce more perithecia with each other than to the other RI group. Some strains produce only few perithecia to one or both tester strains and so could not be confidently assigned to an RI group.

**Clustering analysis of mating compatibility.** Mating compatibility data between the strains of the Wageningen collection produced by ref. [39] was coded into a distance matrix according to the following scheme: mature fruiting bodies produced by both mating partners with or without the presence of ascospore abortion—1; mature fruiting bodies produced by one of the mating partners with or without the presence of ascospore abortion and perithecial abortion observed in the other partner—2; mature fruiting bodies produced by one of the mating partners with or without the presence of ascospore abortion and no perithecia observed on the other partner—3; perithecial abortion observed in both mating partners—4; perithecial abortion observed in one partner of the cross and no perithecia observed in the other partner—5; no perithecia observed on either mating partner—6. The resulting non-symmetric distance matrix (Supplementary Table 3) was transformed into a Gower's dissimilarity matrix using the R v.3.5.1 function daisy from the package cluster v.2.0.7.1. The optimal number of clusters was determined according to maximum average silhouette width as determined by function pam and the Gap statistic[84] also from the cluster package.

**DNA extraction and Illumina paired-end sequencing.** Growth and DNA extraction followed ref. [29]. Briefly, monokaryotic isolates of each strain were grown on $PASM_{0.2}$ plates covered with cellophane, from which 80–100 mg of mycelium were recovered for extraction. Whole genome DNA was extracted with the ZR Fungal/Bacterial Microprep kit (Zymo; www.zymoresearch.com). Library preparation and sequencing with the HiSeq X Illumina technology (150 base pair (bp) long reads) was done at the SNP and SEQ Technology platform (SciLifeLab). For PCR amplification of *het-r* (see below), the mycelium was first homogenized in the Mini-Beadbeater (Biospec Products) for 30 s at 25 Hz previous to extraction.

**Read mapping and variant calling.** We detected adaptors from the Illumina reads using cutadapt v.1.13 (ref. [85]) and removed them using Trimmomatic 0.36 (ref. [86]) with the following options: ILLUMINACLIP:adaptors.fasta:1:30:9 LEADING:20 TRAILING:20 SLIDINGWINDOW:4:20 MINLEN:30. We used the filtered pairs with both forward and reverse reads for downstream analysis. As a reference genome we used the second version of the strain S+ assembly[45] available in Joint Genome Institute MycoCosm website (https://genome.jgi.doe.gov/programs/fungi/index.jsf) as Podan2. Short-read mapping was done with BWA v.0.7.17[87] and PCR duplicates were marked with Picard v.2.19.0 (https://broadinstitute.github.io/picard/). We used HaplotypeCaller of the Genome Analysis Toolkit (GATK) v.4.1.1 (ref. [88]) for indel realignment and variant discovery across all (haploid) samples simultaneously, following the GATK Best Practices recommendations[89]. For subsequent analyses we used only the SNP data (that is, the indels were excluded). Data associated with the mitochondrial scaffold were discarded. VCF file manipulation was done with VCFtools v.0.1.16 (ref. [90]) and BCFtools v.1.9 (ref. [91]). Visual inspection of the mapped reads revealed low levels (<10% of the total depth of coverage, typically 1–3%) of index-hopping in some samples[92]. Because the sequenced strains are haploid and have a high read coverage (>80×), the low level of contaminating reads did not affect base calling.

**PCA of SNP data.** From the SNPs obtained with GATK, we filtered out all sites with the following criteria: QD < 2.0, FS > 60.0, MQ < 40.0, QUAL < 30.0, SOR > 3.0, ReadPosRankSum < −8.0. We also removed all sites that overlapped with repeated elements as detected by RepeatMasker v.4.0.7 (https://www.repeatmasker.org/) with the library of ref. [29]. From the remaining SNPs we only retained those sites without any missing data (hereafter referred to as the 'high-quality' SNP set). We further reduced the high-quality SNP set by filtering out SNPs with minor allele frequency (MAF) <1% and used the packages gdsfmt v.1.20.0 and SNPrelate v.1.18.0 (ref. [93]) in R v.3.5.1 to produce a PCA of the full genome and of each chromosome. A Snakemake pipeline (SNPpop.smk) is available at https://doi.org/10.5281/zenodo.6323682.

This analysis included all samples sequenced, as at least the French strain S also has mating data available to compare it with. However, using only the Wageningen samples leads to similar results. The two samples with intermediate values of PC1 (next to the line at PC1 = 0) in Fig. 2a (Wa60+ and Wa61−), have a *V1* allele (that is, they belong to RI group *RV1*) but are recombinant with the RI group *rV* for the linked sites surrounding the *het-v* locus. Removing these two samples had no effect on downstream analyses.

**Population genetic analyses.** To investigate the genetic diversity of *P. anserina*, we used the R package PopGenome v.2.6.1 (ref. [94]) to estimate the following population genetic parameters: the average pairwise nucleotide diversity within a population ($\pi$) and between two populations ($D_{xy}$) (ref. [95]), Watterson's theta ($\theta_W$), the Tajima's D statistic[96] and the fixation index $F_{st}$ (refs. [97,98]). All analyses were done on windows of 10 kb with steps of 1 kb. Of note, standard VCF files only contain sites that are variable but do not specify sites with missing data that

might or might not be invariant. Since ignoring missing data can artificially alter the parameters estimates (for example, some sites that are invariant in the VCF file could actually just be missing data, which would lower estimates of $\pi$ if the size of the window is not corrected), we used the raw SNP set produced by GATK and we marked explicitly the sites that had either too high or too low coverage as missing data, under the assumption that coverage deviations relate to mis-mapping and repeated elements. To do so, we calculated the depth of coverage distribution of each individual sample and recorded 'bad' sites, which were below the 25% quantile or above the 98.5% quantile (thresholds based on preliminary analyses and manual evaluation of reads mapping to the reference genome) using the package vcfR v.1.8.0 (ref. [99]) and BEDtools v.2.28.0 (ref. [100]). We then intersected regions with 'bad' coverage across all individuals with the RepeatMasker annotation above. All sites overlapping with such ranges were set as missing data explicitly in the VCF file and were excluded from the total window length when calculating $\pi$ and $\theta_W$. Windows with final length <5 kb were set as missing data. To assess the significance of the $F_{st}$ values across the genome we randomly rearranged all samples into groups of the same size as the two RI groups (without replacement) and recalculated $F_{st}$ per window 1,000 times to create a distribution. The location of centromeres was inferred as in ref. [29] on the basis of drops in GC content. A Snakemake pipeline (DiversityStats.smk) is available at https://doi.org/10.5281/zenodo.6323682.

We found that the overall diversity in all sampled strains (Supplementary Table 1) is very low ($\pi = 0.000497$). Still, we restricted the analyses to only the Wageningen strains ($\pi = 0.000492$) to ensure the patterns correspond to a single population.

LD was calculated using the squared allele frequency correlations ($r^2$) (refs. [101,102]). We calculated chromosomal LD decay by combining the methodology of ref. [103] and ref. [104]. Specifically, for each chromosome we randomly sampled $30 \times 50$ kb long windows and extracted all SNPs present in the high-quality SNP with singletons removed. We used VCFtools to calculate $r^2$ for pairs of loci within each window (–hap-r2). We evaluated the decay of LD with distance using a nonlinear regression[105] following equation 1 of ref. [106], which assumes a recombination-drift equilibrium model and low levels of mutation. This equation has a single coefficient $C$ equivalent to the product of the distance between SNPs in bp and the effective population recombination rate $\rho$. This rate, in turn, can be estimated as $\rho = 4N_e c$, where $N_e$ is the effective population size and $c$ is the recombination fraction between sites[105,106]. We obtained least-squares estimates of $\rho$ by fitting the nls function from the base R[104,107]. As additional visualization for the general trend in LD decay, we divided the $r^2$ values into distance intervals of 1 kb and computed the mean $r^2$ for each, plotted with a generalized additive mode smoothing (gam method in ggplot2). To find localized patterns of LD along chromosomes, we calculated $r^2$ between pairs of SNPs as above but with a MAF > 0.02 and subsampled so that no two sites are closer than 1 kb of each other (–thin 1000). We then plotted an LD heatmap within each chromosome, or within and between 1 Mb of chromosomes 2 and 5, centred around *het-r* and *het-v*, respectively.

We found that the LD decay of chromosome 4 is, on average, much slower than in other chromosomes (Supplementary Fig. 3). This seems to be due to the presence of large haplotypes in both subtelomeric regions and around the *het-e* locus, which also show very positive Tajima's D values (Supplementary Fig. 2). When these areas are removed (retaining everything between the positions 1 Mb and 3.5 Mb), the LD decay of chromosome 4 resembles that of other chromosomes (Supplementary Fig. 3).

To assess genome-wide LD with the *het-v* locus, we calculated $r^2$ between *het-v* and other markers, either additional genotyped *het* genes or high-quality biallelic SNPs with a MAF > 0.02 (that is, the variant is present in at least two individuals). The SNPs were processed using the v.1.10.0 of the vcfR package[99]. As expected given random chromosomal segregation, *het-v* in chromosome 5 has very low linkage levels ($r^2 < 0.01$) with genotyped *het* genes in different chromosomes (*het-z* and *het-q*). Still, we found a very strong association with *het-r* in chromosome 2 ($r^2 = 0.926$) as hypothesized due to their interaction but also a moderate association with *het-s* in chromosome 3 ($r^2 = 0.246$). The latter value, however, falls within the distribution of $r^2$ estimates between *het-v* and SNPs along the genome in general (Supplementary Fig. 14b).

**Cloning and characterization of *het-v*.** Deletion strains were obtained by constructing deletion cassettes in which the *Escherichia coli* hygromycin B phosphotransferase (*hph*) or the *Streptomyces noursei* nourseothricin resistance (*nat1*) genes were inserted between ~500–700 bp PCR fragments corresponding to the flanking sequences of the region of interest (position of the flanking regions is given Supplementary Fig. 8). The deletion cassettes were then used to transform *PaKu70::ble* strains derived from the s strain[108], which is of the *V* phenotype. The *rV*, *rV1* and *RV1* strains used for phenotypic testing are backcrosses of the respective alleles into an s strain background[44]. Mutants of the catalytic site residues of HET-Va Y233 were obtained by directed mutagenesis. Homology modelling of the SET and Rubisco domains of HET-Va was done using the HHPred server and Modeller at https://toolkit.tuebingen.mpg.de[109]. Structure models were visualized with CCP4MG[110]. Sequence comparison in Hidden Markov model searches were performed with HHMER[111] (http://hmmer.org/) and HHPred[112].

**Positional cloning of *het-v*.** Since *het-v* was known to be located on the left arm of chromosome 5, we created genetic markers in that region and analysed their linkage to *het-v* (Fig. 3b and Supplementary Fig. 8). The *nat1* marker was introduced in the *idi2* gene (Pa_5_4020) located on the left arm of chromosome 5. The *nwd3* gene (Pa_5_3370) is also located on the left arm of chromosome 5 and displays two alleles, *nwd3-1* and *nwd3-2*, that can be distinguished by the number of WD40 repeats. A *V1 nwd3-2 × V* nwd3-1 *idi2::nat1* cross was set up and 116 homokaryotic progeny were isolated and phenotyped for nourseothricin resistance, *het-v*-incompatibility type in barrage tests and genotyped for repeat length polymorphism at *nwd3* by PCR. Percentages of second division segregation (see ref. [29] for terminology) were 24%, 49% and 51%, respectively, for *idi2*, *nwd3* and *het-r*. Out of 103 progeny, 12 showed recombination between *idi2::nat1* and *V*, a single one showed recombination between *nwd3* and *V* and 11 (10.7%) showed recombination between *idi2* and *nwd3*. The physical distance between *idi2* and *nwd3* is 244 kb, leading to an estimated 1% of recombination for 22.8 kb in that region, suggesting in turn that *V* is located within ~20 kb (~1% recombination) from *nwd3*. A 30 kb region spanning the tentative position of *V* between Pa_5_3300 and Pa_5_3180 was thus amplified and sequenced in a *V1* strain. Polymorphism between *V* and *V1* strains was only detected in Pa_5_3200 gene. Deletion of this gene in the *V* background did not affect *V* incompatibility and thus Pa_5_3200 is not allelic to *V*. Because no candidate for *V* was identified in this region, we set out to obtain a genetic marker that would flank *V* on the centromere distal side. To that end, the *hph* marker was introduced between Pa_5_12920 and Pa_5_12915 located at ~165 kb centromere distal to *nwd3* and ~115 kb from Pa_5_3180 (the marker was termed *hph115*). In 124 progeny of a *V hph115 × V1* no recombinants between *hph115* and *V* were obtained suggesting tight linkage between the marker and *V* (but preventing the determination of the relative order of *nwd3*, *V* and *hph115*).

Rather than further increasing the number of analysed progeny (to determine the relative position of *nwd3*, *V* and *hph115*), we reasoned that the position of the *het-v* gene relative to the *hph115* marker could be determined by deleting the 113 kb sequence spanning the region from Pa_5_3180 to the insertion site of the *hph115* marker. If *V* is centromere proximal to *hph115*, this deletion should affect *het-v* incompatibility. This region containing 28 annotated genes (Pa_5_3170 to Pa_5_12915) was deleted and replaced by the *hph* gene (Supplementary Fig. 8). An *rV* strain was transformed with the deletion cassette and four homokaryotic transformants were obtained. Correct insertion of the deletion cassette was verified by PCR. The transformants deleted for the 113 kb region lost the barrage reaction to *rV1*. Instead, these transformants now produced a barrage reaction to *rV* (Fig. 3c). Thus, these transformants lost the *V* phenotype and at the same time acquired the *V1* phenotype, rather than becoming neutral. The *het-v* locus is thus centromere proximal to *hph115* located within the 113 kb region spanning Pa_5_3170 to Pa_5_12915.

Next, we constructed two smaller deletions (77 kb and 35 kb long), further subdividing the 113 kb region containing *het-v* (Fig. 3b and Supplementary Fig. 8). Deletion of the 77 kb region between Pa_5_12915 and Pa_5_12690 led to transformants displaying the *V1* phenotype (it produced a barrage reaction to *V*). Transformants bearing the 35 kb deletion remained *V* (it produced a barrage reaction to *V1*). The same strategy was repeated with deletions *Δ12915-12820* and *Δ12810-12690* (Supplementary Fig. 8). Transformants carrying the *Δ12810-12690* deletion showed the *V1* phenotype, those carrying the *Δ12915-12820* deletion remained *V*. Attempts were made to further subdivide the region spanning Pa_5_12810 to Pa_5_12690 but the expected *Δ12810-12760* and *Δ12750-12690* deletions could not be obtained even after several trials, suggesting that the chromosomal region is recalcitrant to integration of the deletion cassette.

Our results suggest that the deletion of *V* leads to the *V1* phenotype. Accordingly, earlier genetic investigation failed to recover null mutants for *V*; all mutants recovered for the loss of the *V* phenotype acquired simultaneously the *V1* phenotype[42]. It also shows that the 113 kb region does not contain any essential genes, consistent with the fact that chromosome 5 appears enriched for contingency genes[58].

**Identifying the exact ORFs defining *het-v*.** When examining the 113 kb region in the strain Y (of *V1* compatibility type), we found that five ORFs present in the reference strain S (*V*) were replaced by a cluster of transposable elements of the *discoglosse* (Tc1/mariner-like), *atelopus* (Copia-Ty1) and *crapaud* (Gypsy-Ty3) types (see ref. [45] for nomenclature). To determine which of the five ORF(s) within this region confer the *V* phenotype, we turned to a complementation cloning approach using the *Δ12810-12690 V1* strain as a recipient. The region spanning the five ORFs was amplified in a *V* strain as two PCR fragments, termed *a* (3.4 kb, corresponding to Pa_5_12750, Pa_5_12740 and Pa_5_12730) and *b* (5.7 kb, corresponding to Pa_5_12720 and Pa_5_12710) (Fig. 3b and Supplementary Fig. 8). The *a* and *b* fragments were used to transform a *V1* recipient strain (*Δ12810-12690*). Ten transformants were obtained with fragment *a* but none produced a barrage reaction to *rV1* nor *RV1*. From the 18 transformants obtained with fragment *b*, 14 produced a barrage to *rV1* and *RV1* and not to *rV* (they acquired the *V* phenotype). The remaining four produced no barrage to *rV1* or to *RV1* (still display the *V1* phenotype). Hence, the fragment containing Pa_5_12720 and Pa_5_12710 confers the *V* phenotype to a *V1* strain and thus corresponds to *het-v*.

**Separate roles of Pa_5_12710 (*het-Vb*) and Pa_5_12720 (*het-Va*) in *het-v* incompatibility.** The genes *het-Va* and *het-Vb* were amplified separately and transformed into the same *V1* recipient strain (Supplementary Table 4). Transformants with *het-Va* alone produced a barrage to *RV1* but not to *rV1* nor to *rV*. Transformants with *het-Vb* alone produced a barrage to *rV* but not to *rV1* or *RV1*. We conclude from these experiments that *het-Va* and *het-Vb* together are required for incompatibility to *V1* while *het-Va* alone determines incompatibility to *R*.

**Analysis of sequences of the proteins encoded by *het-Va* and *het-Vb*.**
Gene *het-Va* encodes a 469 amino acid long protein with an N-terminal SET lysine methyltransferase domain (pfam 00856), followed by a Rubisco LSMT substrate-binding domain (pfam 09273) and a C-terminal MYND zinc-binding domain (01753) (Fig. 3d and Supplementary Fig. 9). It is homologous to an *N. crassa* protein that was annotated SET-9 in a systematic description of methyltransferases in that species[113]. Such SET-domain proteins are predicted lysine methyltransferases of histone and non-histone targets. The best BLASTp and HHPred hits of *het-Va* in particular are non-histone methyltransferases (last checked on 8 November 2021). Homology modelling to the human SET6D (pdb 3QXY) and Rubisco LSMT protein from *Pisum sativum* (pdb 2H2J) identified Y233 of Pa_5_12720 as the catalytic tyrosine[114]. Hence, we generated two point mutants, Y233A and Y233F, which confirmed that the methyltransferase activity of this gene is required for *V1/V* incompatibility but dispensable for *R/V* incompatibility (main text). In addition, the two point mutants display a slightly different phenotype: Y233F leads to loss of the barrage reaction to the *V* tester (and in that sense resembles the wild-type allele), whereas Y233A does not, consistent with the fact that the Y to F substitution is more conservative than Y to A. Homologues of *het-Va* (displaying the same domain architecture with a SET/Rubisco/MYND domains) were found only in filamentous fungi in a limited number of species (125 blast hits with e-value <10⁻⁵⁰ in the JGI fungal database with 2027 genome entries on 5 May 2021).

Gene *het-Vb* encodes a 771 amino acid long protein with a N-terminal domain (residue 1 to ~140) showing remote similarity with the pore-forming domain of the mammalian MLKL pseudo-kinase involved in execution of necroptotic cell death[115]. The fungal HeLo and HeLo-like domains involved in the induction of cell death by membrane permeation and in [Het-s]-mediated incompatibility also show homology to MLKL[50]. The N-terminal region (~535–675) of the *het-Vb* encoded protein shows homology to TUDOR domains. TUDOR domains are beta-sheet rich globular domains often repeated in tandem and involved in methyl binding[49]. These low-level homologies were detected using the HHpred server. Gene *het-Vb* homologues were found only in filamentous fungi in a limited number of species (63 blast hits with e-value <10⁻⁵⁰ in the JGI fungal database with 2027 genome entries on 5 May 2021). In a number of species, homologues of both *het-Va* and *het-Vb* were found.

**Genotyping of the *het* genes.** To genotype their allorecognition genes, all samples with Illumina data were assembled de novo using SPAdes v.3.12.0 (ref. [116]) with the *k*-mers 21, 33, 55, 77 and the --careful option. When available for a given strain, we used the long-read assemblies produced in refs. [29,117] instead. We used BLAST searches to extract the genes defining *het-z*[48], *het-s*[118], *het-c*[31] and *het-q*[119] from all assemblies and we manually assigned each sequence to the corresponding functional allele. Notice that genotyping is not yet possible for the remaining *het* genes of *Podospora* (Supplementary Fig. 1). Most variants of *het-d* and *het-e* have not been characterized phenotypically, while the gene(s) responsible for *het-b* have not been cloned yet. We found from sequence alignments of the *P. anserina* stains with long-read data that there is very little sequence variation within the *V* and *V1* allele classes of the *het-v* locus.

The Illumina assembly of genes belonging to the HNWD family[36,59] proved to be unreliable whenever there were more than four WD40 repeats present (on the basis of comparisons with long-read data assemblies and visual inspections of read mapping). Hence, we followed the PCR protocol of ref. [59] to amplify the WD40 domain of *het-r*. We used the primers A_F 5′-GCACCGGTTGGCAGTCTGG-3′ and B_R 5′-CCAGGCCCTTCTCGTGTTAGG-3′ in 25 μl of PCR reaction containing 1× Phusion Green GC Buffer (Thermo Scientific), 200 μM of each dNTPs, 0.5 μM of each primer and 0.02 U μl⁻¹ of Phusion High-Fidelity DNA Polymerase (Thermo Scientific). The cycle condition was as follows: an initial denaturation step at 98 °C for 30 s, then 28 cycles each with denaturation at 98 °C for 10 s, annealing at 68 °C for 30 s and elongation at 72 °C for 30 s, followed by a final extension at 72 °C for 10 min. Amplified products were separated in 1.2% agarose gel containing GelRed (Biotium). GeneRuler Express DNA Ladder and GeneRuler 100 bp Plus DNA Ladder (Thermo Scientific) were used as size markers. The number of repeats in the WD40 domain of each strain was estimated on the basis of the size of the WD40 amplicons assuming that one WD40 repeat corresponds to PCR product of about 126 bp.

Since the order and sequence of the WD40 repeats are also important for allele specificity, we selected a subsample of 23 strains for Sanger sequencing (Supplementary Table 1). PCR products were cleaned with Illustra ExoProStar 1-Step enzyme (GF HealthCare), prepared for sequencing with BigDye Terminator v.3.1 Cycle Sequencing and BigDye XTerminator Purification Kits (Applied Biosystems) and sequenced on an ABI 3730 XL machine using two pairs of nested primers (Supplementary Table 5) from ref. [59]. We used the Geneious 10.1.3 software (https://www.geneious.com) to align the resulting sequences to the long-read data references to confirm allelic identity. The WD40 repeat structure of the strain Wa63 is considered to be representative of *het-R*; all other allelic variants are considered to be *het-r*[36]. The Sanger sequencing results were consistent with the PCR method.

We reasoned that if the RI groups do not mix in nature, we would encounter a strong deficit of recombinant individuals (*rV1*). This is a conservative approach to test for reproductive isolation because: (1) errors in the genotyping method above are more likely to favour an *r* allele since there are many more ways to produce the mutant version (*r*) rather than finding exactly 11 WD40 repeats in a specific order (*R*); and (2) spontaneous vegetative mutations are known to occur, probably breaking *R* into an *r* allele[36,120]. Hence, genotyping biases would increase the number of individuals genotyped as *rV1*.

In the case of the 2017 strains, we genotyped *het-v* by PCR using a primer pair that specifically amplifies a 4,144 bp region of the *V* locus (oliVV1: GTGGGAACGATGGAGGGAGAG and oliV: GCGTTCGTCTCAGCAATCTTGAG) and a primer pair that specifically amplifies a 2,177 bp region of the *V1* locus. We used the Q5 High-Fidelity DNA Pol kit (NEB) according to the manufacturer's recommendation. Annealing temperatures and elongation time were 69 °C and 3 min for the *V* amplification and 66 °C and 2 min for the *V1* amplification.

We searched for both *het-r* and *het-v* in related species of the *P. anserina* species complex[65] with whole genome sequence data available[29,117,121]. We used BLAST searches and manual inspection of synteny to assess presence and orthology. We found that the ortholog of *het-r* is in all species but no strain showed the exact arrangement of 11 repeats found in the *P. anserina R* allele. While these could correspond to the *r* allele, further testing and population sampling is needed. In the case of *het-v*, we only found *het-Va* and *het-Vb* in three other strains, representing two species. As the *het-v* haplotype found in the *P. pauciseta* strain CBS237.71 was almost identical in sequence to that of *P. anserina* (suggesting introgression), we chose four genes flanking the *het-v* haplotype and produced maximum likelihood genealogies of said genes using IQ-TREE v.1.6.8 (refs. [122,123]) with extended model selection (-m MFP) and 100 standard bootstrap pseudoreplicates. We kept only six *P. anserina* strains as representatives of the two RI for clarity within each gene alignment.

To evaluate the presence of the *het-v* genes in species of *Neurospora*, the sequences of *het-Va* and *het-Vb* were used as a BLAST query against the population datasets from refs. [124–126]. The custom script query2haplotype v.1 (available at https://github.com/SLAment/Genomics/blob/master/BLAST/query2haplotype.py) was used with parameters -s 100 -e 1 and -f 100. An absence of hits was taken as evidence that that gene is not present in that genome and those with partial hits were considered fragments. The annotation for the gene models of *N. crassa* and *N. tetrasperma* (FungiDB, https://fungidb.org/fungidb/) place unsupported introns into the sequences to avoid premature stop codons. We assume that the intron structure matches that of *P. anserina* as this holds true for the supported introns and for the distantly related sequences from *Magnaporthe oryzae*. Given this rational, sequences with premature stop codons were assumed to represent putative pseudogenes.

**Genotyping of the *Spok* genes.** The presence or absence of *Spok2*, *Spok3* and *Spok4* was defined first by BLAST searches as for the *het* genes above. Whenever *Spok3* and *Spok4* were present in a sample, either only one gene was assembled (often *Spok4*) or both were fragmented into multiple small scaffolds. The latter was due to tracks of gene conversion between them[29] that complicate the assembly graph. Hence, we mapped the reads of samples with more than one *Spok* to a reference that has all three *Spok* types (Wa87+ from ref. [29]). Manual inspection of the mapped reads allowed us to distinguish the number and type of *Spok* genes, as well as to discard false positives due to low levels of index-hopping.

**Individual-based simulation of the *het-r/het-v* interaction in *P. anserina*.** We use SLiM v.3.3.2 (ref. [60]) to create individual-based simulations that take into account the genetic architecture of the *het-v* and *het-r* genes as well as detailed features of the *Podospora* life cycle. See Supplementary Methods for details. The SLiM and R scripts used to generate and analyse the simulations can be found at https://doi.org/10.5281/zenodo.6323682

**Reporting Summary.** Further information on research design is available in the Nature Research Reporting Summary linked to this article.

## Data availability
Whole genome sequencing was deposited in NCBI SRA under BioProject PRJNA743020. All other data are available as Supplementary Tables.

## Code availability
All custom code is available at Zenodo[127] with the identifier https://doi.org/10.5281/zenodo.6323682

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

## Acknowledgements

We would like to thank M. Grudzinska-Sterno for valuable assistance with laboratory work, as well as R. Debuchy and M. Sicault-Sabourin for assistance in the positional cloning. Next Generation sequencing was possible thanks to the support of the National Genomics Infrastructure/Uppsala Genome Center on massive parallel DNA sequencing. The computations were performed on resources provided by Swedish National Infrastructure for Computing (SNIC) through Uppsala Multidisciplinary Center for Advanced Computational Science (UPPMAX) under the projects SNIC 2017/1-567 and SNIC 2019/8-371. We would also like to thank S. Tusso for useful advice and code peer review. Funding was provided by the European Research Council (ERC) with the grant no. ERC-2014-CoG, project 648143 SpoKiGen (H.J.); the Swedish Research Council with grant no. 2015-04649 (H.J.); the Centre National de la Eecherche Scientifique with recurrent funding to UMR 5095 (S.J.S.); and the Anna Maria Lundin's Travel Grant 2016 and 2017 (S.L.A.-V.).

## Author contributions

S.L.A.-V., A.A.V., H.J., S.J.S. and C.C. were responsible for conceptualization. S.L.A.-V., A.A.V., E.B. and S.D.G. undertook data curation. S.L.A.-V., A.A.V., I.M.-A. and S.J.S. completed the formal analysis. H.J., S.J.S. and S.L.A.-V. acquired funding. S.L.A.-V., A.A.V., A.G.-F., E.B., I.M.-A., S.J.S., S.D.G. and C.C. conducted investigations. S.L.A.-V., A.A.V., A.G.-F., I.M.-A. and C.C. were involved in methodology. S.L.A.-V., A.A.V., S.J.S. and H.J. undertook project administration. A.J.M.D. obtained resources. S.L.A.-V. was responsible for software. H.J., S.J.S., A.J.M.D. and M.L. undertook supervision. S.L.A.-V., I.M.-A. and S.J.S. conducted visualization. S.L.A.-V. wrote the orginal draft manuscript. S.L.A.-V., A.A.V., I.M.-A., M.L., H.J., S.J.S. and A.J.M.D. reviewed and edited the manuscript.

## Funding

## Competing interests

The authors declare no competing interests.

## Additional information

**Correspondence and requests for materials** should be addressed to S. Lorena Ament-Velásquez or Hanna Johannesson.

# Reporting Summary

## Statistics

For all statistical analyses, confirm that the following items are present in the figure legend, table legend, main text, or Methods section.

| n/a | Confirmed | |
|---|---|---|
| ☐ | ☒ | The exact sample size (*n*) for each experimental group/condition, given as a discrete number and unit of measurement |
| ☒ | ☐ | A statement on whether measurements were taken from distinct samples or whether the same sample was measured repeatedly |
| ☐ | ☒ | The statistical test(s) used AND whether they are one- or two-sided<br>*Only common tests should be described solely by name; describe more complex techniques in the Methods section.* |
| ☒ | ☐ | A description of all covariates tested |
| ☒ | ☐ | A description of any assumptions or corrections, such as tests of normality and adjustment for multiple comparisons |
| ☐ | ☒ | A full description of the statistical parameters including central tendency (e.g. means) or other basic estimates (e.g. regression coefficient) AND variation (e.g. standard deviation) or associated estimates of uncertainty (e.g. confidence intervals) |
| ☐ | ☒ | For null hypothesis testing, the test statistic (e.g. *F*, *t*, *r*) with confidence intervals, effect sizes, degrees of freedom and *P* value noted<br>*Give P values as exact values whenever suitable.* |
| ☒ | ☐ | For Bayesian analysis, information on the choice of priors and Markov chain Monte Carlo settings |
| ☒ | ☐ | For hierarchical and complex designs, identification of the appropriate level for tests and full reporting of outcomes |
| ☒ | ☐ | Estimates of effect sizes (e.g. Cohen's *d*, Pearson's *r*), indicating how they were calculated |

*Our web collection on statistics for biologists contains articles on many of the points above.*

## Software and code

Policy information about availability of computer code

| Data collection | Most of the data used was generated in this study, with the exception of the raw mating success data (as explained in the Main text and Methods) and the genome sequence of a few strains from the same population published previously (accession numbers available in the Supplementary Table 1). The new strains collected in 2016 and 2017 were deposited in the collection of Wageningen University & Research. |
|---|---|
| Data analysis | All code (in python, Snakemake and R) used for the bioinformatic analyses is available at the GitHub repository https://github.com/johannessonlab/HetVPaper |

For manuscripts utilizing custom algorithms or software that are central to the research but not yet described in published literature, software must be made available to editors and reviewers. We strongly encourage code deposition in a community repository (e.g. GitHub). See the Nature Portfolio guidelines for submitting code & software for further information.

## Data

Policy information about availability of data

All manuscripts must include a data availability statement. This statement should provide the following information, where applicable:
- Accession codes, unique identifiers, or web links for publicly available datasets
- A description of any restrictions on data availability
- For clinical datasets or third party data, please ensure that the statement adheres to our policy

Whole genome sequencing was deposited in NCBI under BioProject PRJNA743020. All other data is available as supplementary tables. The associated code is available in the GitHub repository https://github.com/johannessonlab/HetVPaper.

# Field-specific reporting

Please select the one below that is the best fit for your research. If you are not sure, read the appropriate sections before making your selection.

☐ Life sciences    ☐ Behavioural & social sciences    ☒ Ecological, evolutionary & environmental sciences

For a reference copy of the document with all sections, see nature.com/documents/nr-reporting-summary-flat.pdf

# Ecological, evolutionary & environmental sciences study design

All studies must disclose on these points even when the disclosure is negative.

| | |
|---|---|
| Study description | In this study we used a Dutch collection of 106 strains from the fungus *Podospora anserina* to assess the connection between vegetative and sexual incompatibility and the effects on speciation. Using whole genome Illumina sequencing, we characterized the genetic diversity of this population and detected signatures of balancing selection for known vegetative incompatibility loci (het genes). Using previously published mating success data for the same population, we identified two reproductively isolated groups defined by the antagonistic interaction of two het genes: het-r and het-v. While het-r was already known, here we characterize het-v through positional and complementation cloning, as well as site-directed mutagenesis. We used substrate information of the collection, plus additional sampling of 68 strains to determined that the two reproductively isolated groups co-occur in their known substrate. Finally, we used individual-based simulations to gain insights into the evolution of this system. |
| Research sample | The Wageningen Collection consist of strains isolated around the city of Wageningen, The Netherlands, as described in Materials and Methods. Sequenced strains were collected between 1991 to 2016. Additional strains were sampled in 2017 but not sequenced. This fungus has a "pseudo-homothallic" reproductive system, meaning that the mycelium contains two types of haploid nuclei, one of each mating type, which allows for self-fertilization. It occasionally produces haploid ascospores, which were used for all the analyses. Here, a strain correspond to a single ascospore extracted from a fruiting body present in herbivore dung, as well as all its descendants derived by selfing. The Wageningen collection has been studied extensively for the evolution of meiotic drive, het genes, and senescence. This includes the original raw data of mating success found in the PhD thesis of Marjin van der Gaag, and which we re-analyzed here as described in Materials and Methods. |
| Sampling strategy | We attempted to include all the strains of the Wageningen Collection, but the spores of a few strains failed to germinate or had failed sequencing and were not included in the project. We also attempted to include available strains from other parts of the world (France, Switzerland, Canada, and Argentina), but very few strains of this and related species are available elsewhere. |
| Data collection | Sequencing, sampling, phenotyping and genotyping is described in the Materials and Methods section. Sampling of new strains in 2016 and 2017 was done by the authors SLAV, SDG, EB, and AJMD. Coding of mating success data was done by AAV. Phenotyping to assign strains from 2017 to the reproductively isolated groups was done by AAV and SLAV. Cloning, mutagenesis, and genotyping of het-v was done by AGF and CC. Preparation of cultures for sequencing was done by EB. All simulation analyses were done by IMA. |
| Timing and spatial scale | Samples were collected between 1991 and 2017 in association with several PhD projects from Wageningen University & Research and Uppsala University, as well as the course of Advanced Genetics imparted by AJMD. Isolating new strains is a laborious and specialized effort that is strongly subject to chance (e.g. mass mortality of the local rabbit population in some years). |
| Data exclusions | No data was excluded. Whenever the analyses included strains other than those sampled in Wageningen, this is specified in the Materials and Methods or in the Main text. |
| Reproducibility | The main bioinformatic analyses were done using the workflow manager Snakemake available in the GitHub repository of the paper. Otherwise, they can be reproduced from stand-alone R scripts also in the repository. The procedure to locate and characterize het-v, including failed attempts, are described in the Materials and Methods. |
| Randomization | All Wageningen samples were included in all analyses, as the objective was to characterize the population. The samples are only randomly assigned to groups to calculate the distribution of Fst values expected by chance. |
| Blinding | Blinding was not necessary as all the fungal strains are indistinguishable and there is no phenotypic trait associated to the reproductively isolated groups (other than the vegetative and sexual incompatibility itself) |

Did the study involve field work?    ☒ Yes    ☐ No

## Field work, collection and transport

| | |
|---|---|
| Field conditions | Field work was done in grassland-like areas around Wageningen, the Netherlands, and strain isolation was done as described in Materials and Methods. Only the source herbivore for the dung was recorded. |
| Location | Wageningen, the Netherlands. The coordinates for the 2017 samples are provided in the Supplementary Table 2. |
| Access & import/export | As both Sweden and the Netherlands are parties of the Nagoya protocol, but have no Access and Benefit-Sharing legislation, a declaration of due diligence was not required. |
| Disturbance | The samples taken (pieces of dung from horses, cows, sheep and rabbit) were small and found mostly in agricultural areas or near roads. |

# Reporting for specific materials, systems and methods

We require information from authors about some types of materials, experimental systems and methods used in many studies. Here, indicate whether each material, system or method listed is relevant to your study. If you are not sure if a list item applies to your research, read the appropriate section before selecting a response.

## Materials & experimental systems

| n/a | Involved in the study |
|-----|----------------------|
| ☒ | Antibodies |
| ☒ | Eukaryotic cell lines |
| ☒ | Palaeontology and archaeology |
| ☐ | ☒ Animals and other organisms |
| ☒ | Human research participants |
| ☒ | Clinical data |
| ☒ | Dual use research of concern |

## Methods

| n/a | Involved in the study |
|-----|----------------------|
| ☒ | ChIP-seq |
| ☒ | Flow cytometry |
| ☒ | MRI-based neuroimaging |

## Animals and other organisms

Policy information about studies involving animals; ARRIVE guidelines recommended for reporting animal research

| | |
|---|---|
| Laboratory animals | The data associated to all Podospora strains used in this study is available in Supplementary tables 1 and 2. |
| Wild animals | The study involved fungal strains isolated from the wild as described above and in the Materials and Methods. |
| Field-collected samples | The maintenance and storage of the fungal strains is described in Materials and Methods. |
| Ethics oversight | For experimental work with Podospora anserina, no ethical approval or guidance was required. |

Note that full information on the approval of the study protocol must also be provided in the manuscript.

