## [Peer Review File. · Nature Ecology & Evolution]

Peer Review Information

Journal: Nature Ecology & Evolution

Manuscript Title: Allorecognition genes drive reproductive isolation in *Podospora anserina*

Corresponding author name(s): S. Lorena Ament-Velásquez, Hanna Johannesson

Editorial Notes:

Reviewer Comments & Decisions:

Decision Letter, initial version:

8th October 2021

Dear Dr Ament-Velásquez,

Your manuscript entitled "Allorecognition genes drive reproductive isolation in *Podospora anserina*" has now been seen by three reviewers, whose comments are attached. The reviewers have raised a number of concerns which will need to be addressed before we can offer publication in Nature Ecology & Evolution. We will therefore need to see your responses to the criticisms raised and to some editorial concerns, along with a revised manuscript, before we can reach a final decision regarding publication.

In particular, Reviewers #1 and #3 have suggested some additional experiments that they and we (the editors) hope will contribute to further strengthening your study. My apologies that some of the concerns raised by Reviewer #1 appear to be due to how our manuscript tracking system presented the files to the reviewers; I will also explain this to Reviewer #1.

We therefore invite you to revise your manuscript taking into account all reviewer and editor comments. Please highlight all changes in the manuscript text file.

* If you have not done so already please begin to revise your manuscript so that it conforms to our Article format instructions at <http://www.nature.com/natecolevol/info/final-submission>. Refer also to any guidelines provided in this letter.

[REDACTED]

Nature Ecology & Evolution is committed to improving transparency in authorship. As part of our efforts in this direction, we are now requesting that all authors identified as 'corresponding author' on published papers create and link their Open Researcher and Contributor Identifier (ORCID) with their account on the Manuscript Tracking System (MTS), prior to acceptance. ORCID helps the scientific community achieve unambiguous attribution of all scholarly contributions. You can create and link your ORCID from the home page of the MTS by clicking on 'Modify my Springer Nature account'. For more information please visit please visit www.springernature.com/orcid.

[REDACTED]

Reviewer expertise:

2Reviewer #1: genomics and molecular biology, non-self recognition in fungi

Reviewer #2: genetic basis of incompatibility, speciation

Reviewer #3: genomics, incompatibility and speciation

Reviewers' comments:

Reviewer #1 (Remarks to the Author):

Review of "Allorecognition genes drive reproductive isolation in *Podospora anserina*" by Ament-Velásquez et al., describes a mostly unbiased population genomics approach to define genes involved in sexual incompatibility in the self-fertile species, *Podospora anserina*. Using this method on a single population from Wageningen, The Netherlands, the authors identify the previously molecularly characterized het-r and genetically characterized het-v as potential candidates; these two loci have been previously determined to regulate heterokaryon incompatibility in an allelic (het-v) and non-allelic (het-r/het-v) manner. The authors molecularly characterized het-v and determine that one population (of the het-V1 genotype) lack two linked methyltransferase genes (PaMt1 and PaMt2) present in the het-V genotype. The methyltransferase activity is important for het-V1/V allelic incompatibility, but is dispensable for non-allelic R/V incompatibility; the other domains of PaMt1/PaMt2 were not evaluated (MLKL/TUDOR).

1. The supplemental information provided in the PDF was incomprehensible (Supplementary Materials and Methods, Figs. S1 to S17, Tables S1 to S6, References (63–118)). Only one figure was provided in the PDF, no figure legends, Tables had no titles or numbers, a bunch of lines filling pages was included. Therefore, none of the supplemental data cited in the text could be properly evaluated.
2. More information should be included on the life cycle of *P. anserina*—how vegetative incompatibility relates to sexual sterility is not clear. How does allelic incompatibility occur during sexual reproduction? Were strain only crossed on plates, where hyphal fusion and vegetative incompatibility play a role or where they also fertilized by spermatia of the opposite mating type? A discussion of this and inclusion of additional data would clarify this point.
3. It would be most useful to know when the block in mating compatibility occurred in the het-r/v incompatibility versus the het-V1/V incompatibility, and both together. Fertilization? Ascospore formation? Ascospore germination?
4. Is non-allelic incompatibility between het-r/het-v similar to allelic incompatibility mediated by het-V1/V. This aspect could be made clearer.
5. It is not clear if point mutants lacking the methyltransferase activity of PaMt1/PaMt2 affect sterility; apparently only incompatibility was monitored. This experiment should be easy to do and would inform mechanisms of sexual sterility. It would be interesting to know if other regions of PaMt1/PaMt2 (MLKL/Tudor) also conferred differential incompatibility.

36. Some of the terminology is difficult to follow. A simple diagram outlining the role of het-r/het-v in vegetative and sexual incompatibility with regards to the life cycle of *P. anserina* would be useful—it could be put in supplemental and would really help the non-fungal expert.

7. Maybe Table 4: truly dikaryotic? Or is heterokaryotic a more accurate term? Referring to ascospores or hyphal culture? And by monokaryotic, is the meaning homokaryotic? Also, footnote cut off, as with other tables.....

8. A bit more information on spore killers is needed to understand the potential relationship between het-v and SPOK and reproductive isolation/speciation.

9. The role of het-r/het-v and het-VI/het-V in sexual incompatibility and RI versus genetic differences at other het loci, which also lead to sexual incompatibility could be made clearer. Is the expectation that other *P. anserina* populations might be different?

Reviewer #2 (Remarks to the Author):

The authors sequence and analyze the genome data from more than 100 *P. anserina* strains collected around Wageningen in the Netherlands. By integrating the mating success data, they discover that these strains can be separated into two reproductively isolated (RI) groups and the division is correlated with the het-r/v incompatibility. Experiments are carried out to map and validate that the PaMt1 and PaMt2 genes in the het-v locus contribute to the het-r/v incompatibility. They further examine another set of samples collected in 2017 and confirm that individuals from two RI groups coexist in natural habitats but hybrids are rare. A simulation model reveals that balancing selection and selfing are essential parameters for the invasion of incompatible mutants. Lastly, they show that the het-r/v genes also exist in some related species. This is a beautiful and complete work combining different approaches to understand how reproductive isolation evolves between different populations. Although some aspects of the heterokaryon incompatibility have been investigated or discussed in previous studies, it does not diminish the novelty and impact of the current work. I only have a few specific comments.

Specific comments:

- 1) What are the possible driving forces underlying the balancing selection for the incompatible alleles? Since balancing selection plays a crucial role in the formation of RI here and is also observed in many het loci, this issue needs to be carefully discussed.
- 2) The references of Fig. S1 are missing. Also, do we know why most sexual incompatibility observed in *P. anserina* is asymmetric? It will be nice to provide some explanations or speculations for general readers.
- 3) In Fig. 1A, why doesn't the MAT locus show the balancing selection signature?
- 4) Line 71: "we sampled 68 new genetically different strains..." It is unclear what "genetically different" means here and how the authors can be sure that they are completely different

4(genetically?) from previously collected strains.

5) Line 73: the logic behind the spore killing assay needs to be clearly explained. In rarely outcrossing populations, don't we expect spore killing to happen more often between different isolates?

6) In Fig. S4, it will be nice to show all other het loci with the balancing selection signature.

7) In Fig. S6, the light blue F_{st} obtained from permutations is almost invisible. Changing to a different color (e.g., pink) may help.

8) Reference 26: please correct the source.

9) Line 314: this reference is missing in the reference list.

Reviewer #3 (Remarks to the Author):

This manuscript describes new insights into a fascinating non-self recognition system in the fungus *P. anserina*. It spans the full circle from investigating genetic diversity in a local collection of wild isolates, functional tests of unlinked non-self recognition loci, and confirmation with samples from the wild that there is opportunity for outcrossing, with non-self recognition events ("barrage" directly observable in the wild samples. I particularly liked the straight-forward approach of clustering samples by whole-genome relatedness, and to then look for highly differentiated regions as candidates for harboring incompatibility loci. A further strength is the longitudinal data over more than two decades. The genetics is still lacking, but the necessary experiments should be easily within reach.

My main concern is that the authors consider V1 a simple deletion, but it actually is a replacement of V with a cluster of TEs. It is therefore unclear whether the interaction between V and V1 is correctly considered as a hemizygous effect (i.e., interaction between V and absence of V); rather, it seems at least as likely that it is an interaction between the two PaMT genes constituting V and the TE cluster. This could be tested by making a deletion of this TE cluster. The converse experiment is a cross between RV and the deletion strain, which seems to be missing. (Apologies if I overlooked it!) Does the deletion strain behave like RV1, as predicted by the authors?

Such an interaction does not seem all that unlikely, given that PaMT1 appears to encode a cytosine methylase. Along these lines, the authors could say a bit more about what kind of methylase it is, and they could say in general more about cytosine methylation in this species (please cite the work from Bewick et al., 2019!).

Minor comments:

The lethality of RV/RV was described in 1974. I would have liked to see this confirmed with the cloned V locus, i.e., transformation into an rV1 strain and subsequent crosses with an RV strain.

The SLiM simulation is a nice touch, but could one not have made use of the actual data to estimate outcrossing rates more directly (residual runs of heterozygosity in wild strains)? (Are there too few polymorphisms for this?)

The first part (unbiased Tajima's D scans), while interesting, distracts from the overall story. I suggest to either exclude it, or better, to move the independent genotyping data of het-r from Fig 4A to Fig 1, and to add the same information for het-d. Similarly, the paragraphs on Spok2/het-v linkage and het-r and het-v in other species are not very compelling.

More care should be taken in preparing the figures (consistency in fonts etc.).

The PaMT1/PaMT2 nomenclature is confusing, since the two proteins are unrelated.

*****END*****

Author Rebuttal to Initial commentsResponse to Reviewers for “Allorecognition genes drive reproductive isolation in *Podospora anserina*”

Response: We thank the editor and reviewers for their time and effort to review our manuscript.

In summary, these are the main major changes we have done to the manuscript:

- In response to questions from Reviewer 1, we have carried out a new experiment on the fertility effects of the Pa_5_12720 point-mutants, and added a new paragraph, new supplementary figure (Fig. 11) and supplementary table (Table S5) to the manuscript, outlining the results.
- Also in response to comments from Reviewer 1, we have added a new summary figure (Fig. 4) illustrating the vegetative, prezygotic and postzygotic effects of *het-r/v*. This figure is also influenced by a comment of Reviewer 2.
- We rewrote the final two sections of the results based on comments from Reviewers 1 and 3.

Additional minor changes to the manuscript include:

- Based on a comment of Reviewer 3, we changed the names of the genes from *PaMt1* and *PaMt2* to *het-Va* and *het-Vb*, respectively.
- The addition of a diagram (Figure S12) showing the segregation of *het-r* and *het-v* in the spores, to make clear that the actual proportion of spores that die from the *R/V* incompatibility reaction is unknown.
- We changed all the references to parts of figures from “A”, “B”, etc. to “a”, “b”, etc.
- We also changed the spelling to British English, and edited the text for clarity.

We list each comment below and respond to them one-by-one:

Reviewers' comments:

Reviewer #1 (Remarks to the Author):

Review of “ Allorecognition genes drive reproductive isolation in *Podospora anserina*” by Ament-Velásquez et al., describes a mostly unbiased population genomics approach to define genes involved in sexual incompatibility in the self-fertile species, *Podospora anserina*. Using this method on a single population from Wageningen, The Netherlands, the authors identify the previously molecularly characterized *het-r* and genetically

characterized *het-v* as potential candidates; these two loci have been previously determined to regulate heterokaryon incompatibility in an allelic (*het-v*) and non-allelic (*het-r/het-v*) manner. The authors molecularly characterized *het-v* and determine that one population (of the *het-V1* genotype) lack two linked methyltransferase genes (*PaMt1* and *PaMt2*) present in the *het-V* genotype. The methyltransferase activity is important for *het-V1/V* allelic incompatibility, but is dispensable for non-allelic *R/V* incompatibility; the other domains of *PaMt1/PaMt2* were not evaluated (*MLKL/TUDOR*).

1. The supplemental information provided in the PDF was incomprehensible (Supplementary Materials and Methods, Figs. S1 to S17, Tables S1 to S6, References (63–118)). Only one figure was provided in the PDF, no figure legends, Tables had no titles or numbers, a bunch of lines filling pages was included. Therefore, none of the supplemental data cited in the text could be properly evaluated.

Response: We thank the reviewer for taking the time and effort to try to understand our paper, even without having access to the supplementary materials. Unfortunately, this seems to be due to an issue with the manuscript tracking system, as explained by the Editor. Hopefully the files will be available now.

2. More information should be included on the life cycle of *P. anserina*—how vegetative incompatibility relates to sexual sterility is not clear. How does allelic incompatibility occur during sexual reproduction? Were strain only crossed on plates, where hyphal fusion and vegetative incompatibility play a role or where they also fertilized by spermatia of the opposite mating type? A discussion of this and inclusion of additional data would clarify this point.

Response: We expect that this concern can partly be explained by the lack of access to supplementary information. In the original submission we showed the full life cycle in Fig. S15, and this should now be accessible. Nevertheless, to address this and other comments, we have now included an additional main figure (now Fig. 4), which indicates where exactly in the life cycles the effects of the *het-r/v* system take place, based on multiple previous observations using classical genetics and experiments. In particular, in response to the specific questions of the reviewer, it has been reported that allelic *V/V1* sexual incompatibility occurs -as for other *het*-systems in *P. anserina*-, post-fertilization and at a prezygotic stage, and corresponds to an abortive fertilization preventing further development of the female organ (Bernet 1965). In allelic *V/V1* sexual incompatibility female organs remain competent for fertilization with compatible spermatia (= microconidia) in contrast to *R/V* non-allelic sexual incompatibility that damages female organs and precludes their subsequent fertilization with compatible spermatia (Labarère et al. 1974). This information is now given in the new Fig. 4, and references are added.

In our and previous experiments, crosses were performed both on plates and by fertilization with spermatia suspensions. This is indicated in the Methods section “Culture and crossing conditions”; it can also be seen by comparing the Fig. 3E (confrontation) and Fig. S10 (spermatization). Please note, however, that when crosses are performed on plates, fertilization

also proceeds from the interaction of spermatia and female organs and is not dependent on vegetative hyphae that would act as a male gamete. If this were the case, that is, if hyphal fertilization plays a significant quantitative role in crosses on plates, one would expect all incompatibility systems to affect sexual reproduction and in a symmetric way, which is not the case. We added a short note in the caption of Fig. 3, and small cartoons in the sexual compatibility diagram to emphasize the fact that fertilization occurs between the protoperithecium and microconidia.

- Bernet, J. Mode d'action des gènes de barrage et relation entre l'incompatibilité cellulaire et l'incompatibilité sexuelle chez le *Podospora anserina*. *Annales des sciences naturelles Botanique* 6, 611–768 (1965).

- Labarère, J., Bègueret, J. & Bernet, J. Incompatibility in *Podospora anserina*: comparative properties of the antagonistic cytoplasmic factors of a nonallelic system. *Journal of Bacteriology* 120, 854–860 (1974).

3. It would be most useful to know when the block in mating compatibility occurred in the het-r/v incompatibility versus the het-V1/V incompatibility, and both together. Fertilization? Ascospore formation? Ascospore germination?

Response: These questions are also addressed by the new Fig. 4. As mentioned above, the block both for *RV* and *VV1* incompatibility occurs after fertilization and at a prezygotic stage, with the difference that *R* female organs are damaged after aborted fertilization with incompatible *V* spermatia while *V* female organs remain competent for fertilization. In both cases, in the fraction of protoperithecia in which fertilization proceeds, ascospore formation is normal and so is germination. Cell death occurs about 15 hours after germination in *RV* progeny and is suppressed at 32°C (Labarère et al. 1974; Bidard et al. 2013). In dikaryotic *VV1* progeny (that is formed when *V* shows second division segregation during meiosis), there is no manifestation of incompatibility -in contrast to other *het*-systems in that species- and those dikaryons grow normally and are fertile. In other words, the *V* allele (the presence of the two genes) is dominant over *V1* (absence of the genes), as hinted by Fig. 3c and its caption.

- Bidard, F., Clavé, C. & Saupe, S. J. The transcriptional response to nonself in the fungus *Podospora anserina*. *G3 Genes|Genomes|Genetics* 3, 1015–1030 (2013).

- Labarère, J., Bègueret, J. & Bernet, J. Incompatibility in *Podospora anserina*: comparative properties of the antagonistic cytoplasmic factors of a nonallelic system. *Journal of Bacteriology* 120, 854–860 (1974).

4. Is non-allelic incompatibility between het-r/het-v similar to allelic incompatibility mediated by het-V1/V. This aspect could be made clearer.

Response: Hopefully the new figure also partially addresses this point. To explain further, the non-allelic and allelic interactions are similar in that they all cause a barrage reaction (Bernet 1992), and this is true for all known *het* genes in *Podospora* (allelic and non-allelic). Phenotypically, Labarère et al. (1974) described the *RV* interaction in great detail, and the *VV1*

interaction is discussed in the thesis of J. Labarère (1978). Having said that, the two incompatibility systems also differ in many aspects. Specifically, they show different extragenic and chemical suppressors and show a difference in temperature sensitivity. Also, as already mentioned, the fertility defect in *R/V* and *V/V1* sexual incompatibility is also different. All of these aspects have been extensively explained in cited work. Still, in order to clarify this, we recapitulate some of these aspects in a new paragraph associated with the new Fig. 4 (starting at line 347) and its caption.

The molecular insights provided by the present work further illustrate these differences as we show that the methyltransferase activity is dispensable for one of the incompatibility reactions (*RV*), and that *RV* incompatibility relies on just one of the genes (*Pa_5_12720*) while *V/V1* incompatibility requires both (Fig. 3d).

- Bernet, J. In *Podospora anserina*, protoplasmic incompatibility genes are involved in cell death control via multiple gene interactions, *Heredity* 68:79-87 (1992).
- Labarère, J., Bègueret, J. & Bernet, J. Incompatibility in *Podospora anserina*: comparative properties of the antagonistic cytoplasmic factors of a nonallelic system. *Journal of Bacteriology* 120, 854–860 (1974).
- Labarère, J. L'incompatibilité protoplasmique chez le champignon *Podospora anserina*: étude génétique et biochimique, relation avec certains aspects du développement et de la morphogénèse. (Université de Bordeaux II, 1978).

5. It is not clear if point mutants lacking the methyltransferase activity of PaMt1/PaMt2 affect sterility; apparently only incompatibility was monitored. This experiment should be easy to do and would inform mechanisms of sexual sterility. It would be interesting to know if other regions of PaMt1/PaMt2 (MLKL/Tudor) also conferred differential incompatibility.

Response: As suggested by the reviewer, we performed a new experiment, in which we verified that the mutants of the methyltransferase abolish the sterility effects of the allelic *het-v* system (*V/V1*), as expected from the vegetative incompatibility interactions. However, we found that one of the mutants (Y233A) suppresses the *R/V* sexual incompatibility but not the vegetative incompatibility (the mutant Y233F leads to no sexual or vegetative *R/V* incompatibility). That means that it is possible to decouple the sexual and vegetative phenotypes, to a degree. Thus, this experiment gave new and interesting results. We thank the reviewer for the suggestion. We added these new results starting line 332.

Unlike for the methyltransferase, we lack a clear target to test for the effects of the MLKL/Tudor domains, and that would probably require considerable further testing to define relevant mutants. We felt that this would not add significantly to the main message of the paper, and instead would be a good follow up on the molecular details of the *het-v* function. Hence, for this question, we chose to limit ourselves to the current data.

6. Some of the terminology is difficult to follow. A simple diagram outlining the role of *het-r/het-v* in vegetative and sexual incompatibility with regards to the life cycle of *P. anserina* would be useful—it could be put in supplemental and would really help the non-fungal expert.

Response: This suggestion resulted in the new Fig. 4. We decided to show it as a main figure as we think it contributes greatly to the clarity of the paper.

7. Maybe Table 4: truly dikaryotic? Or is heterokaryotic a more accurate term? Referring to ascospores or hyphal culture? And by monokaryotic, is the meaning homokaryotic? Also, footnote cut off, as with other tables.....

Response: We assume the Reviewer refers to Table S2. The terms "Dikaryotic" and "monokaryotic" are often used interchangeably with "heterokaryotic" and "homokaryotic" in the *Podospora* literature. We choose to use dikaryotic and monokaryotic to emphasize the fact that there are only one or two types of nuclei in a given mycelium, as defined by the mating type. The missing footnote of the column "Original culture" in the original submission says:

"[2] Only dikaryotic spores were isolated from the fruiting bodies, but likely some cultures sectorized into monokaryotic areas after germination, as revealed by mating type phenotyping. This resulted in the loss of one nucleus."

8. A bit more information on spore killers is needed to understand the potential relationship between *het-v* and SPOK and reproductive isolation/speciation.

Response: As two reviewers had difficulty with the spore killers parts, we removed the topic from the introduction and instead added a few explanatory sentences at line 102. We also fully rewrote the section "The *het-v* locus is physically close to a meiotic drive gene" to explain our ideas. In brief, both *Spok* genes and the *RV* incompatibility system experience fitness penalties at low frequencies, making invasion difficult. Their physical proximity suggests that they may have aided each other in overcoming this initial barrier to reach higher frequencies in the population.

9. The role of *het-r/het-v* ad *het-VI/het-V* in sexual incompatibility and RI versus genetic differences at other *het* loci, which also lead to sexual incompatibility could be made clearer. Is the expectation that other *P. anserina* populations might be different?

Response: Other *het*-interactions (*C/D*, *C/E* and *Z1/Z2*) lead to pleiotropic effects in crosses (Bernet 1967). In all cases, the prezygotic effects are asymmetric, which means that in each incompatible combination the cross remains fully fertile in one direction. *C/D*, *C/E* and *Z1/Z2* sterility occurs only when the *D*, *E* and *Z1* alleles respectively are harboured by the female parent. Because *C/D* and *C/E* incompatibility is non-allelic and each locus has multiple alleles, sexual incompatibility can be additive. For instance, a *C1E1* x *C2E2* cross is nearly totally sterile. Yet, there are numerous wild-isolates that show neutral *het-d* and *het-e* alleles (Bernet

1976). Therefore, presumably these systems do not lead to such a strong reproductive isolation effect as *het-r* and *het-v*, because strains with neutral *het-d* and *het-e* alleles can bridge gene flow between incompatible isolates. For *R/V* and *V/V1* incompatibility, *rV1* strains could play the same role as this genotype shows normal fertility—at least in one direction—to both *rV* and *RV1*, but this is precisely the genotype that is absent from the population. We attempted at explaining this point in the text starting line 556.

Our expectation would indeed be that if allelic distributions of *het* genes are very different in other populations the effect on reproductive isolation could vary. For example, if a population shows few or no neutral *het-e* and *het-d* alleles it might be expected that *C/D* and *C/E* incompatibility might also lead to strong gene flow barriers. Partial evidence, however, suggests that at least the French strain collection collected by Georges Rizet and Jean Bernet in France in the 1950's and the Wageningen population have comparable *het*-allele distribution and constitution.

Reviewer #2 (Remarks to the Author):

The authors sequence and analyze the genome data from more than 100 *P. anserina* strains collected around Wageningen in the Netherlands. By integrating the mating success data, they discover that these strains can be separated into two reproductively isolated (RI) groups and the division is correlated with the *het-r/v* incompatibility. Experiments are carried out to map and validate that the *PaMt1* and *PaMt2* genes in the *het-v* locus contribute to the *het-r/v* incompatibility. They further examine another set of samples collected in 2017 and confirm that individuals from two RI groups coexist in natural habitats but hybrids are rare. A simulation model reveals that balancing selection and selfing are essential parameters for the invasion of incompatible mutants. Lastly, they show that the *het-r/v* genes also exist in some related species. This is a beautiful and complete work combining different approaches to understand how reproductive isolation evolves between different populations. Although some aspects of the heterokaryon incompatibility have been investigated or discussed in previous studies, it does not diminish the novelty and impact of the current work. I only have a few specific comments.

Response: We thank the reviewer for the positive comments and constructive questions!

Specific comments:

1) What are the possible driving forces underlying the balance selection for the incompatible alleles? Since balancing selection plays a crucial role in the formation of RI here and is also observed in many *het* loci, this issue needs to be carefully discussed.

Response: We modified the introduction to discuss this further. We originally had:

"However, fusion between different individuals can be deleterious since it allows for the transmission of viruses, defective plasmids, and selfish nuclei 17–21. It is believed that the primary function of the *het* genes is to avoid such risks, as successful fusion is only possible if individuals are compatible at all of their *het* genes, otherwise triggering regulated cell death of the fused cell."

Now we also refer to another interesting theory:

"It has also been proposed that some *het* genes are involved in pathogen recognition, and that the vegetative incompatibility is a secondary by-product of their evolution 23,24. In that sense, the *het* genes can be seen as analogous to the innate immune system of animals and plants 24."

As suggested by another reviewer, we have included a new summary figure with a diagram of the effects from the *het-r/v* system at several points in the life-cycle (new Fig. 4). To emphasize the point about the adaptiveness of the *het* genes, we have included a cartoon of a virus that fails to infect a new strain when barrage formation occurs in Fig. 4a.

2) The references of Fig. S1 are missing. Also, do we know why most sexual incompatibility observed in *P. anserina* is asymmetric? It will be nice to provide some explanations or speculations for general readers.

Response: We added the relevant references to the caption of Fig. S1.

We added an explanation of the asymmetry in the *het-r/v* system starting in the new line 347. Basically, asymmetry has been connected to either the amount or the mobility of the incompatibility factors (Bernet 1965, Labarère 1974). The HET-R protein (an NLR) is proposed to be too low in amount or poorly diffusible so that it is essentially absent from the male gamete (microconidia). Therefore, no adverse effects occur at fertilization in the male *R* x female *V* context. Notice that the other systems with an asymmetric relationship also involve an NLR: one of the genes from the *het-z* locus (Heller et al. 2018) and the *het-e* and *het-d* genes (Paoletti et al. 2007) are different types of NLRs.

- Bernet, J. Mode d'action des gènes de barrage et relation entre l'incompatibilité cellulaire et l'incompatibilité sexuelle chez le *Podospora anserina*. *Annales des sciences naturelles Botanique* 6, 611–768 (1965).

- Labarère, J., Bègueret, J. & Bernet, J. Incompatibility in *Podospora anserina*: comparative properties of the antagonistic cytoplasmic factors of a nonallelic system. *Journal of Bacteriology* 120, 854–860 (1974).

- Heller, J., Clavé, C., Gladieux, P., Saupe, S. J. & Glass, N. L. NLR surveillance of essential SEC-9 SNARE proteins induces programmed cell death upon allorecognition in filamentous fungi. *Proceedings of the National Academy of Sciences* 115, E2292–E2301 (2018).

- Paoletti, M., Saupe, S. J. & Clavé, C. Genesis of a Fungal Non-Self Recognition Repertoire. *PLoS ONE* 2, e283 (2007).

3) In Fig. 1A, why doesn't the MAT locus show the balancing selection signature?

Response: It does, in fact, as it has positive Tajima's D value. However, the signal is weak because most SNPs within the MAT locus itself were removed by the filtering pipeline. This is due to the fact that the MAT "alleles" (idiomorphs) are extremely divergent and contain different genes, so only the reads from samples with the same MAT as the reference map to that region. Thus, the small positive peak around the MAT locus comes from linked variants. In addition, we arbitrarily chose one mating type or the other when isolating haploids for sequencing. In reality, all strains originally have the two mating types (because they are dikaryotic and pseudo-homothallic) and by definition they will have perfectly balanced MAT allele frequencies.

To avoid confusion, we added a note in the caption of that figure, saying:

"Loci with highly divergent alleles (e.g. MAT) and repetitive regions (e.g. WD40 repeats of HNWD genes) were filtered out by the variant-calling pipeline, but linked variants can still show signals of balancing selection."

4) Line 71: "we sampled 68 new genetically different strains..." It is unclear what "genetically different" means here and how the authors can be sure that they are completely different (genetically?) from previously collected strains.

Response: The methods to identify these strains as genetically different are described in the Methods section "Reproductively isolated (RI) group phenotypic assignment":

"Multiple strains could be isolated from the same dung sample. To determine if these isolates represented unique strains or separate isolations of the same clone, all isolates from the same dung sample were confronted against each other under standard conditions on HPM medium. Formation of a barrage between strains was used to indicate that the isolates represented unique genets. If no barrage was formed, isolates were considered to represent the same strain."

To help the reader, we pointed to the Methods section at the end of that sentence (now line 101).

5) Line 73: the logic behind the spore killing assay needs to be clearly explained. In rarely outcrossing populations, don't we expect spore killing to happen more often between different isolates?

Response: We changed the original sentence:

"Additionally, we sampled 68 new genetically different strains in 2017 around Wageningen (table S2; see also (26)) and investigated the occurrence of spore killing, a phenotypic

expression of meiotic drive in *Podospora* that can only occur as a result of an outcrossing event (25)."

For:

"Additionally, we sampled 68 new genetically different strains in 2017 around Wageningen (table S2; see Methods) and investigated the occurrence of spore killing, a phenotypic expression of meiotic drive in fungi. Previous studies have shown that *P. anserina* has a number of meiotic drivers (selfish genetic elements that cause segregation distortion) from the *Spok* gene family 29,30. Mating between an individual with a meiotic driver and an individual without it results in the abortion of the spores that did not inherit the meiotic driver. Hence, observing spore killing is a direct indication of an outcrossing event."

It has been shown with population genetic models that spore killers will have a harder time invading populations with high selfing rates (Martinossi-Allibert et al. 2021). This happens because selfing decreases the opportunity of having heterozygotes crosses, which is where meiotic drive happens. So, in fact, in rarely outcrossing populations spore killers should be even rarer, and so most isolates would not be killers (or alternatively, a lucky killer would be fixed already, everybody would be a killer and there would be no heterozygous crosses). Observing spore killing simply indicates that there was an outcrossing event between a strain with the killer and a naive (non-killer) strain in the history of that meiosis. Since the active killers are in relatively low frequencies, it means that there must be sufficient outcrossing for us to encounter the unlikely event of a killer crossing with a non-killer.

Notice that we say "active" killers, to emphasize the fact that the spore-killing observed in this populations is mostly derived from the action of *Spok3* and *Spok4*, which are in low frequencies. Most strains already contain *Spok2*, so most spore-killing events are not due to this gene. This dynamic happens because all the *Spok* genes can kill each other (they have no epistatic interactions). This complicated system is described in Vogan et al. (2019), but for our purposes here it's sufficient to know that spore-killing happens at low frequencies and it is the direct result of outcrossing.

- Martinossi-Allibert et al. 2021. Invasion and maintenance of meiotic drivers in populations of ascomycete fungi, *Evolution* 75: 1150-1169.

- Vogan, A. A. et al. Combinations of *Spok* genes create multiple meiotic drivers in *Podospora*. *eLife* 8, e46454 (2019).

6) In Fig. S4, it will be nice to show all other het loci with the balancing selection signature.

Response: Unfortunately, the only genes from which we can produce these plots are already in Fig. 1b and Fig. S4. We have no genotyping information of *het-d* and *het-e*, as these genes are multiallelic and have not been characterized in the same way as *het-r*. The genes for *het-b* have

not been discovered yet. We added an explanation of this in the Methods section "Genotyping of the *het* genes", lines 1028-1031.

7) In Fig. S6, the light blue Fst obtained from permutations is almost invisible. Changing to a different color (e.g., pink) may help.

Response: We changed it as suggested.

8) Reference 26: please correct the source.

Response: Mistake corrected.

9) Line 314: this reference is missing in the reference list.

Response: Mistake fixed.

Reviewer #3 (Remarks to the Author):

This manuscript describes new insights into a fascinating non-self recognition system in the fungus *P. anserina*. It spans the full circle from investigating genetic diversity in a local collection of wild isolates, functional tests of unlinked non-self recognition loci, and confirmation with samples from the wild that there is opportunity for outcrossing, with non-self recognition events ("barrage" directly observable in the wild samples. I particularly liked the straight-forward approach of clustering samples by whole-genome relatedness, and to then look for highly differentiated regions as candidates for harboring incompatibility loci. A further strength is the longitudinal data over more than two decades. The genetics is still lacking, but the necessary experiments should be easily within reach.

Response: We thank the reviewer for the positive feedback.

My main concern is that the authors consider V1 a simple deletion, but it actually is a replacement of V with a cluster of TEs. It is therefore unclear whether the interaction between V and V1 is correctly considered as a hemizygous effect (i.e., interaction between V and absence of V); rather, it seems at least as likely that it is an interaction between the two PaMT genes constituting V and the TE cluster. This could be tested by making a deletion of this TE cluster.

Response: The reviewer correctly points out that the *V1* allele is a cluster of TEs. However, the deletions of the genes in the *V* haplotype fully recapitulate the *V1* phenotype of incompatibility, showing that it is a hemizygous effect. This is explained in the "*het-r* and *het-v* are the causal loci behind the vegetative and sexual incompatibility" section:

"Deletion strains lacking this region lost the barrage reaction to an *rV1* strain, as expected if *het-v* locates there. Notably, while losing the *V* phenotype, these deletion strains simultaneously acquired the *V1* phenotype (rather than showing a neutral incompatibility phenotype) (Fig. 3c). In other words, deleting the region with the *V* allele resulted in vegetative incompatibility to *V* and compatibility to *V1*."

Please notice that if the TEs were necessary, the absence of the two genes (now called *het-Va* and *het-Vb*) would not be enough to produce the incompatibility phenotype with an *rV* strain (as shown in Fig. 3c). In other words, the deletion of *V* would be compatible with both *V* and *V1* (a neutral phenotype), but that is not the case. (See also below the question about methyltransferases, where we explain that *het-Va* (before *PaMt1*) is not a cytosine methyltransferase).

As we had all this information in the manuscript, we did not make any changes to accommodate this concern.

The converse experiment is a cross between RV and the deletion strain, which seems to be missing. (Apologies if I overlooked it!) Does the deletion strain behave like RV1, as predicted by the authors?

Response: The strain *RV* is self-incompatible and lethal (see Fig. 3a), and thus it is not possible to use it for crossings under normal conditions. However, the deletion strain we made ($\Delta 12915-123170$, originally an *rV* strain but with *het-v* removed) behaves like an *rV1* strain, as expected from a hemizygous effect. Moreover, the cross of the deletion strain with an *RV1* strain confirms that also fertility is recovered (Fig. 3e center and Fig. S10).

As for the previous point, we had the information already in the manuscript and therefore have not made any changes to respond to the question.

Such an interaction does not seem all that unlikely, given that PaMT1 appears to encode a cytosine methylase. Along these lines, the authors could say a bit more about what kind of methylase it is, and they could say in general more about cytosine methylation in this species (please cite the work from Bewick et al., 2019!).

Response: The Pa_5_12720 product displays a SET domain involved in lysine and histidine methylation and found in histone but also non-histone methyltransferases. It is orthologous to a protein annotated SET-9 in *Neurospora crassa* but for which no functional data are available. Please note that Pa_5_12720 does not encode a cytosine methylase. Since the reference by Bewick et al. 2019 specifically deals with fungal cytosine methylases we feel it is not directly relevant and chose not to cite it.

However, to avoid confusion, we added "PaMt1 encodes a predicted lysine or histidine methyltransferase (...)" in the new line 319 to be clear in this regard. We further added in the methods the sentence (new line 978-980):

"The best BLASTp and HHPred hits of *het-Va* in particular are non-histone methyltransferases (last checked on November 8th 2021)."

Minor comments:

The lethality of RV/RV was described in 1974. I would have liked to see this confirmed with the cloned V locus, i.e., transformation into an rV1 strain and subsequent crosses with an RV strain.

Response: As the *RV* strain is self-incompatible and dies after germination, we cannot use it for crosses. (A dikaryotic *RV/RV* strain cannot be produced normally because the parents themselves are lethal). But please note that transformation assays and barrage essays with the cloned *V* locus indicate lethality indirectly through the vegetative effects. Specifically, an *rV1* strain transformed with *PaMt1* (Pa_5_12720) (either wild or mutant) produces a barrage reaction with a *RV1* tester (Table S4). It also recapitulates the sexual effects in the new Fig. S11.

The SLiM simulation is a nice touch, but could one not have made use of the actual data to estimate outcrossing rates more directly (residual runs of heterozygosity in wild strains)? (Are there too few polymorphisms for this?)

Response: All the strains were sequenced as haploids (monokaryons), as mentioned in the Methods section "Fungal material", and at the beginning of the section "*Podospora anserina* has extremely low genetic diversity but strong signals of balancing selection". Hence, it was not possible to assess heterozygosity of the wild strains, unfortunately, and we were not able to estimate outcrossing rates with that method. In addition, the strains that were isolated before 2016 were selfed in the lab an unknown number of times, and hence we wouldn't know the real heterozygosity of those strains even if the dikaryons were sequenced. This is also explained in the Materials and Methods section "Fungal material". Thus, we chose to explore a range of values for the parameters in the simulations.

The first part (unbiased Tajima's D scans), while interesting, distracts from the overall story. I suggest to either exclude it, or better, to move the independent genotyping data of *het-r* from Fig 4A to Fig 1, and to add the same information for *het-d*.

Response: While we agree that it can be seen as a lot of extra information, we feel that it is easier to understand the evolution of the *het-r/v* system once the general characteristics of the population are described (i.e. low genetic diversity and high, but not absolute, levels of selfing). We also chose to introduce the Tajima's D scans right from the start to present evidence that *het* genes in general are under balancing selection, a key aspect of the system (and the

simulations) that confirms the adaptive value of the *het* genes. To accommodate the point raised, we rephrased the beginning of the second paragraph of the Results to improve the flow in this regard.

It is possible to move the *het-r* genotyping data to the Fig. 1, but we find it hard to justify the effort of genotyping this difficult gene before concluding that the interaction with *het-v* is the cause of reproductive isolation (and which was the real order of events during our research). Moreover, multiple *het* genes are present in chromosome 3 and they all have SNPs linked that indicate balancing selection, so it is very illustrative. In the case of *het-r* and *het-d* in chromosome 2, there are no linked variants with that signature (Fig. S2), so then it does not illustrate the point of balancing selection (even though we later confirm that this signature is there for *het-r* once we had the PCR genotyping). Unfortunately, we cannot add the information for *het-d* because it does not exist. Like for *het-r* (Chevanne et al. 2009), *het-d* requires a very careful characterization of the phenotypic allelic categories, along with PCR or long-read sequencing of the population. The fact that *het-d* is multiallelic makes this particularly challenging. (This is also true for *het-e*). We added a sentence in the methods section “Genotyping of the *het* genes” to make clear what *het* genes are not available.

- Chevanne, D. et al. Identification of the *het-r* vegetative incompatibility gene of *Podospora anserina* as a member of the fast evolving HNWD gene family. *Current Genetics* 55, 93–102 (2009).

Similarly, the paragraphs on *Spok2/het-v* linkage and *het-r* and *het-v* in other species are not very compelling.

Response: To address this comment, we attempted to clarify further our results and interpretations by restructuring the two relevant sections: “The *het-v* locus is physically close to a meiotic drive gene” and “The *het-r* and *het-v* loci are present in related species”. Hopefully the sections are not clearer.

More care should be taken in preparing the figures (consistency in fonts etc.).

Response: We tried to make the fonts more consistent across all figures, and kept the labels of panels in lower case (a, b, c, etc).

The *PaMT1/PaMT2* nomenclature is confusing, since the two proteins are unrelated.

Response: Based on this comment we changed the names of the genes of *PaMt1* and *PaMt2* to *het-Va* and *het-Vb*, respectively. Although the names remain similar, now they refer to their locus and not to their domains, so hopefully they do not give the impression of being homologous. These names also follow more closely the nomenclature of other *het* genes in *Podospora*. For example, the proteins of the *het-q* are called HET-Q1 and HET-Q2. So now the proteins of *het-v* are named HET-Va and HET-Vb.

Decision Letter, first revision:

31st January 2022

Dear Dr. Ament-Velásquez,

Thank you for submitting your revised manuscript "Allorecognition genes drive reproductive isolation in *Podospora anserina*" (NATECOLEVOL-210814360A). It has now been seen again by the original reviewers and their comments are below. The reviewers find that the paper has improved in revision, and therefore we'll be happy in principle to publish it in Nature Ecology & Evolution, pending minor revisions to satisfy the reviewers' final requests and to comply with our editorial and formatting guidelines.

[REDACTED]

Reviewer #1 (Remarks to the Author):

This is a nicely and thoroughly revised manuscript, and the authors have addressed all my concerns. I have only a few minor suggestions.,

Cite fig. S9 in line 221. Also, note mutated catalytic residue more clearly in the figure or figure legend.

Rename supplemental tables so that, if downloaded by readers, it will be obvious what is contained in them.

Fig. 4a, replace x (which means cross) with + (for heterokaryon confrontation).

S15 figure legend "trichogyne"

S16: define "a" and "b" in legend.

20Reviewer #2 (Remarks to the Author):

The authors have addressed all my questions in the revised manuscript.

Reviewer #3 (Remarks to the Author):

I am rather disappointed by the authors' rebuttal. Their answer is essentially that this reviewer did not understand the complex genetics of the system. I am willing to concede this point, but I consider myself a card-carrying geneticist, and therefore I am surprised that the authors have made essentially no attempt to better explain their system.

For example, in response to my first comment, they did not even amend the statement in the text that the deletions not only prove that the deleted region is responsible for the V phenotype, but also that the sequences that remain in V1 are likely not responsible for the incompatibility with rV. I note that formally it cannot be excluded that they do contribute, since there could be an interaction between the TE sequences in strain Y V1 and other sequences in the strain. If I am not mistaken, the deletion strains are not completely equivalent to the Y V1 strain, since they are not isogenic. In other words, what speaks against the TE sequences being present somewhere else in the genome of the deletion strain?

Some of the difficulty in understanding the system almost certainly comes from the authors' sloppy use of language. For example, they state that "the RV strain is lethal because of self-incompatibility". Lethal means "causing death". I do hope that exposure to the RV strain is not deadly for the experimenter. Rather, what the authors seem to mean is that a consequence of the self-incompatibility of the RV genotype is that it is inviable.

Similarly, relevant information is missing in the main text. Here is one relevant example: "As the het-v locus was genetically mapped to the left arm of chromosome 5, we introduced genetic markers in that region and analysed their linkage to the vegetative incompatibility phenotype of het-v (fig. S8)." First, the genetic mapping data are presumably from ref. 40, but this is not referenced here. Second, what are the genetic markers, how were they introduced, and in which strain were they introduced? Third, what was the cross (strain background and R/V genotype) in which the mapping was performed? Perhaps some of this is obvious or trivial for a fungal geneticist, but it is not for a non-fungal geneticist, who has only ever worked with endogenous, as opposed to introduced, markers. Many of the details can be found in the methods, but it would greatly help if they would at least be cursorily mentioned in the main text.

A practical suggestion is to indicate, for example, the background of the different strains and not only their genotype at R/V, and to include the deletion strains in the diagram in Fig. 3a.

I would strongly encourage the authors to work harder to make their wording more rigorous and their presentation more accessible.

21Our ref: NATECOLEVOL-210814360A

2nd February 2022

Dear Dr. Ament-Velásquez,

Thank you for your patience as we've prepared the guidelines for final submission of your Nature Ecology & Evolution manuscript, "Allorecognition genes drive reproductive isolation in *Podospora anserina*" (NATECOLEVOL-210814360A). Please carefully follow the step-by-step instructions provided in the attached file, and add a response in each row of the table to indicate the changes that you have made. Please also check and comment on any additional marked-up edits we have proposed within the text. Ensuring that each point is addressed will help to ensure that your revised manuscript can be swiftly handed over to our production team.

****We would like to start working on your revised paper, with all of the requested files and forms, as soon as possible (preferably within two weeks). Please get in contact with us immediately if you anticipate it taking more than two weeks to submit these revised files.****

In recognition of the time and expertise our reviewers provide to Nature Ecology & Evolution's editorial process, we would like to formally acknowledge their contribution to the external peer review of your manuscript entitled "Allorecognition genes drive reproductive isolation in *Podospora anserina*". For those reviewers who give their assent, we will be publishing their names alongside the published article.

Nature Ecology & Evolution offers a Transparent Peer Review option for new original research

22manuscripts submitted after December 1st, 2019. As part of this initiative, we encourage our authors to support increased transparency into the peer review process by agreeing to have the reviewer comments, author rebuttal letters, and editorial decision letters published as a Supplementary item. When you submit your final files please clearly state in your cover letter whether or not you would like to participate in this initiative. Please note that failure to state your preference will result in delays in accepting your manuscript for publication.

Cover suggestions

As you prepare your final files we encourage you to consider whether you have any images or illustrations that may be appropriate for use on the cover of Nature Ecology & Evolution.

Nature Ecology & Evolution has now transitioned to a unified Rights Collection system which will allow our Author Services team to quickly and easily collect the rights and permissions required to publish your work. Approximately 10 days after your paper is formally accepted, you will receive an email in providing you with a link to complete the grant of rights. If your paper is eligible for Open Access, our Author Services team will also be in touch regarding any additional information that may be required to arrange payment for your article.

Please note that *Nature Ecology & Evolution* is a Transformative Journal (TJ). Authors may publish their research with us through the traditional subscription access route or make their paper immediately open access through payment of an article-processing charge (APC). Authors will not be required to make a final decision about access to their article until it has been accepted. [Find out more about Transformative Journals](https://www.springernature.com/gp/open-research/transformative-journals)

Authors may need to take specific actions to achieve [compliance](https://www.springernature.com/gp/open-research/funding/policy-compliance-faqs) with funder and institutional open access mandates. For submissions from January 2021, if your research is supported by a funder that requires immediate open access (e.g. according to [Plan S](https://www.springernature.com/gp/open-research/plan-s-compliance)

principles) then you should select the gold OA route, and we will direct you to the compliant route where possible. For authors selecting the subscription publication route our standard licensing terms will need to be accepted, including our [self-archiving policies](https://www.springernature.com/gp/open-research/policies/journal-policies). Those standard licensing terms will supersede any other terms that the author or any third party may assert apply to any version of the manuscript.

[REDACTED]

[REDACTED]

Reviewer #1:

Remarks to the Author:

This is a nicely and thoroughly revised manuscript, and the authors have addressed all my concerns. I have only a few minor suggestions.,

Cite fig. S9 in line 221. Also, note mutated catalytic residue more clearly in the figure or figure legend.

Rename supplemental tables so that, if downloaded by readers, it will be obvious what is contained in them.

Fig. 4a, replace x (which means cross) with + (for heterokaryon confrontation).

S15 figure legend "trichogyne"

S16: define "a" and "b" in legend.

Reviewer #2:

Remarks to the Author:

24The authors have addressed all my questions in the revised manuscript.

Reviewer #3:

Remarks to the Author:

I am rather disappointed by the authors' rebuttal. Their answer is essentially that this reviewer did not understand the complex genetics of the system. I am willing to concede this point, but I consider myself a card-carrying geneticist, and therefore I am surprised that the authors have made essentially no attempt to better explain their system.

For example, in response to my first comment, they did not even amend the statement in the text that the deletions not only prove that the deleted region is responsible for the V phenotype, but also that the sequences that remain in V1 are likely not responsible for the incompatibility with rV. I note that formally it cannot be excluded that they do contribute, since there could be an interaction between the TE sequences in strain Y V1 and other sequences in the strain. If I am not mistaken, the deletion strains are not completely equivalent to the Y V1 strain, since they are not isogenic. In other words, what speaks against the TE sequences being present somewhere else in the genome of the deletion strain?

Some of the difficulty in understanding the system almost certainly comes from the authors' sloppy use of language. For example, they state that "the RV strain is lethal because of self-incompatibility". Lethal means "causing death". I do hope that exposure to the RV strain is not deadly for the experimenter. Rather, what the authors seem to mean is that a consequence of the self-incompatibility of the RV genotype is that it is inviable.

Similarly, relevant information is missing in the main text. Here is one relevant example: "As the het-v locus was genetically mapped to the left arm of chromosome 5, we introduced genetic markers in that region and analysed their linkage to the vegetative incompatibility phenotype of het-v (fig. S8)." First, the genetic mapping data are presumably from ref. 40, but this is not referenced here. Second, what are the genetic markers, how were they introduced, and in which strain were they introduced? Third, what was the cross (strain background and R/V genotype) in which the mapping was performed? Perhaps some of this is obvious or trivial for a fungal geneticist, but it is not for a non-fungal geneticist, who has only ever worked with endogenous, as opposed to introduced, markers. Many of the details can be found in the methods, but it would greatly help if they would at least be cursorily mentioned in the main text.

A practical suggestion is to indicate, for example, the background of the different strains and not only their genotype at R/V, and to include the deletion strains in the diagram in Fig. 3a.

I would strongly encourage the authors to work harder to make their wording more rigorous and their presentation more accessible.

Author Rebuttal, first revision:Second response to Reviewers for “Allorecognition genes drive reproductive isolation in *Podospora anserina*”

We thank once again all the reviewers and the Editors for the time invested in our manuscript. Please notice that we changed the format of the main text (e.g., we removed the main figures) to comply with the editorial checks for the final submission.

Comments of the reviewers are in blue and our answers in black.

Reviewer #1

This is a nicely and thoroughly revised manuscript, and the authors have addressed all my concerns. I have only a few minor suggestions.

We thank again the reviewer for the useful feedback!

Cite fig. S9 in line 221. Also, note mutated catalytic residue more clearly in the figure or figure legend.

Done, now in line 186.

Rename supplemental tables so that, if downloaded by readers, it will be obvious what is contained in them.

Now all tables are in a single file (as per instructions of the Editorial check), with a different table in each tab. We added a caption in each tab for clarity.

Fig. 4a, replace x (which means cross) with + (for heterokaryon confrontation).

Thank you for pointing this out. We replaced it with a “/” to be consistent with the terminology of the main text.

S15 figure legend “trichogyne”

Fixed.

S16: define “a” and “b” in legend.

Done: we moved the description of the figure in the Methods to the caption.

Reviewer #2

The authors have addressed all my questions in the revised manuscript.

We are very grateful for the feedback.

Reviewer #3

I am rather disappointed by the authors' rebuttal. Their answer is essentially that this reviewer did not understand the complex genetics of the system. I am willing to concede this point, but I consider myself a card-carrying geneticist, and therefore I am surprised that the authors have made essentially no attempt to better explain their system.

The remaining criticism of reviewer #3 relate to the section describing molecular genetic identification of *het-v*. We apologize if this section and our rebuttal was not satisfactory to the reviewer and will attempt to rephrase and clarify our response and modify the manuscript where needed. Based on their comments, we agree that more explicit treatment of the strains' origin and phenotype was necessary and made several small clarifying changes in the text accordingly.

For example, in response to my first comment, they did not even amend the statement in the text that the deletions not only prove that the deleted region is responsible for the *V* phenotype, but also that the sequences that remain in *V1* are likely not responsible for the incompatibility with *rV*. I note that formally it cannot be excluded that they do contribute, since there could be an interaction between the TE sequences in strain *Y V1* and other sequences in the strain. If I am not mistaken, the deletion strains are not completely equivalent to the *Y V1* strain, since they are not isogenic. In other words, what speaks against the TE sequences being present somewhere else in the genome of the deletion strain?

We respectfully acknowledge the reviewer's comment but confess that we are unable to fully grasp why the reviewer thinks that this alternative model is not invalidated by the experimental data.

We suspect one source of confusion might have been a misleading usage of the word "deletion" in line 174, that we now replace by "absent". Notice no modifications were done in the wild strain *Y* at all, and likewise the *Y* strain was not used for our phenotypic assays. It was all done in a lab strain (PaKu70) background that originally had a *V* phenotype. This was re-emphasized in the methods section "Cloning and characterization of *het-v*". The description of the locus in the strain *Y* is there as a comparison with the wildtype *V1* and to narrow down the number of candidate genes.

In their initial report, the reviewer questioned our conclusion that the *V* and *V1* phenotype are determined by the presence or the absence of the Pa_5_12710 and Pa_5_12720 genes. The reviewer points out rightly that strain *Y* differs from the *V* reference strain not only by absence of 5 genes (including Pa_5_12710 and Pa_5_12720) but also by the presence at the same locus of several TEs. This, they state, raises the possibility that the TEs of the *Y* strain participate (perhaps as *trans*-acting target sequences) to the *V1* phenotype. In essence, if we understand the reviewer's model correctly, incompatibility could be caused by the interaction of the *het-Va* and *het-Vb* gene products with these TE sequences found in *V1*. The genetics of *VV1* would indeed be consistent with such a model IF there was no further information than the formal genetics of the system and the sequence comparison between the *V* reference strains and the *Y* strain (a wild-type strain of *V1* phenotype). However, we

did rule out this hypothesis because the V-derived deletion strains (either $\Delta 113$, $\Delta 77$ or $\Delta 38$) that never had this TE cluster region show incompatibility to the original strain without the deletion (*V*). In other words, this comparison is totally isogenic except for the *het-v* locus. Therefore, incompatibility cannot be caused by the interaction of *het-Va* and *het-Vb* gene products with the TE sequences. The reviewer then in their second report, we think, raises the possibility that the same or equivalent TE sequences (participating in *V/V1* incompatibility) could be present in the deletion strain elsewhere in the genome, not at the locus they occupy in strain *Y*. But perhaps the reviewer did not realize at that point and also in their first report that the deletion was performed in a *V* strain and not in the *Y* strain. These tentative *V1*-determining TE elements would be absent in the *V* strain (from which the deletion strain is derived). Independently of the deletion experiment, also please note that this hypothesis (presence of *V1*-determining TEs elsewhere in the genome) becomes inconsistent with the genetics of *V/V1*, because in that model *V/V1* would behave as a non-allelic system with Pa_5_12710 and Pa_5_12720 defining *V* and the TEs defining an unlinked *V1*.

To summarize, we report that a *V* strain in which these two *het-Va* and *het-Vb* genes (among others) are deleted acquires the *V1* phenotype, and that re-introduction of these two genes in the deletion strain restores the *V* phenotype. In other words, strains differing only by the presence and absence of *het-Va* and *het-Vb* genes display the alternate *V* and *V1* phenotypes. To us this demonstrates that the *V* and *V1* phenotypes are determined by presence or absence of the *het-Va* and *het-Vb* genes alone (and not by additional linked or unlinked polymorphism between the *V* and *V1* genotypes).

To further avoid any confusion, we have also modified the first two paragraphs of the Results section “*het-v* encodes two genes that cause incompatibilities” to emphasize that the deletion was done in an *rV* strain and that the comparisons are isogenic. We further added a couple of sentences starting line 170 to clarify that the TE cluster is not responsible for the *V1* phenotype:

*“Consistent with the deletion strain experiment, these five genes are contained within the $\Delta 12810$ - 12690 area. The *V1* phenotype of the deletion strains, in turn, indicates that the cluster of transposable elements at the wild-type *V1* locus is not required for the *V/V1* incompatibility.”*

We also changed our formulations in the text (line 174-178). We originally had:

*“To determine which ORF(s) within the region deleted in the strain *Y* confer the *V* phenotype, we turned to a complementation cloning approach using the $\Delta 12810$ - 12690 *V1* strain as a recipient. Our results showed that insertion of the genes Pa_5_12710 and Pa_5_12720 confer the *V* phenotype to a *V1* strain, and that these two genes thus correspond to *het-v*.”*

Here the $\Delta 12810$ - 12690 strain (that was described in lines 158-159 as being genetically derived from a *V* strain) is designated *V1* based on its phenotype. We have reformulated this sentence and also restated specifically that the deletions were performed in a *V* background in line 158. It now reads:

*“To determine which of the five ORF(s) within the region absent in the strain *Y* confer the *V* phenotype, we turned to a complementation cloning approach using the *rV* $\Delta 12810$ - 12690 strain (which displays the *V1* phenotype) as a recipient. Our results showed that insertion of the genes Pa_5_12710 and Pa_5_12720 confer the recipient strain with a *V* phenotype, and that these two genes thus correspond to *het-v*.”*

In addition, in the title of table S4 reporting the results of these experiments the recipient was designated by its phenotype as *rV1*. We have modified the title to stress that it is the deletion strain derived from *rV*.

Some of the difficulty in understanding the system almost certainly comes from the authors' sloppy use of language. For example, they state that "the RV strain is lethal because of self-incompatibility". Lethal means "causing death". I do hope that exposure to the RV strain is not deadly for the experimenter. Rather, what the authors seem to mean is that a consequence of the self-incompatibility of the RV genotype is that it is inviable.

We inspected the occurrences of "lethal" in the manuscript and response letter. We apologize for the incorrect use of the word in the response letter. Fortunately, in the manuscript, all nine occurrences of the term refer to the RV genotype (which indeed can be defined as "causing death") and are thus correct uses of the term. We regret that our wording was somewhat more colloquial in the response letter.

As described above, we tried to be more explicit with the phenotype and genotypes of the strains.

Similarly, relevant information is missing in the main text. Here is one relevant example: "As the *het-v* locus was genetically mapped to the left arm of chromosome 5, we introduced genetic markers in that region and analysed their linkage to the vegetative incompatibility phenotype of *het-v* (fig. S8)." First, the genetic mapping data are presumably from ref. 40, but this is not referenced here. Second, what are the genetic markers, how were they introduced, and in which strain were they introduced? Third, what was the cross (strain background and R/V genotype) in which the mapping was performed? Perhaps some of this is obvious or trivial for a fungal geneticist, but it is not for a non-fungal geneticist, who has only ever worked with endogenous, as opposed to introduced, markers. Many of the details can be found in the methods, but it would greatly help if they would at least be cursorily mentioned in the main text.

The reviewer is correct to state that the mapping information is absent from the main text. This was part of an effort to keep the manuscript length to an acceptable level. We would like, however, to point out that the procedure is described in full in the Methods section. The mapping was done by us, it is not from the reference 40 (see Methods section "Positional cloning of *het-v*"). To help the reader, we now point to the Methods section in the sentence reporting the genetic mapping (line 157). We feel that a partial reiteration of the description of the mapping procedure in the main text would significantly increase the length of this paragraph but still fall short of informing the interested reader. We therefore would like to retain this line of description to a single, fully phrased occurrence in the method section.

The deletion strain (where the mapping was done) corresponds to the PaKu70 strain from reference 106 as stated in the Methods. But given the confusion with the deletion mutants, we agree that it is necessary to be more explicit with the origin of the strains. Hence, we added that this strain is derived from the strain *s* (lowercase *s*, not to be confused with the capital *S*), and that the tester strains have the focal alleles of *het-r* *het-v* introduced by backcrossing into this same strain background (now in section "Cloning and characterization of *het-v*").

A practical suggestion is to indicate, for example, the background of the different strains and not only their genotype at R/V, and to include the deletion strains in the diagram in Fig. 3a.

All genetic and molecular genetic experiments described in the paper are done with reference *R*, *r*, *V* and *V1* strains (and derivatives of the reference *rV* obtained by gene deletion or gene insertions). These reference strains have been previously isogenized through a large number of backcrosses to the strain *s* (as mentioned above and in the Methods) and are isogenic except for the *het-v* locus. Therefore, these strains have in fact all the same genetic background and can be designated by their *het-r het-v*-genotype only.

If possible, we would like to avoid including the deletion strain in the diagram of Fig. 3a because this diagram is intended as a basic presentation of the phenotyping properties of natural *het-r het-v* genotypes. If one was to include the deletion strain, other constructed derivative strains would have to be included and the figure would become a heterogeneous assembly of premises (based on early genetic studies) and the experimental data presented in the paper.

I would strongly encourage the authors to work harder to make their wording more rigorous and their presentation more accessible.

Hopefully the presented changes corrected the sources of confusion.

Final Decision Letter:

15th March 2022

Dear Dr Ament-Velásquez,

We are pleased to inform you that your Article entitled "Allorecognition genes drive reproductive isolation in *Podospora anserina*", has now been accepted for publication in Nature Ecology & Evolution.

Over the next few weeks, your paper will be copyedited to ensure that it conforms to Nature Ecology and Evolution style. Once your paper is typeset, you will receive an email with a link to choose the appropriate publishing options for your paper and our Author Services team will be in touch regarding any additional information that may be required

You will not receive your proofs until the publishing agreement has been received through our system

Due to the importance of these deadlines, we ask you please us know now whether you will be difficult to contact over the next month. If this is the case, we ask you provide us with the contact information (email, phone and fax) of someone who will be able to check the proofs on your behalf, and who will be available to address any last-minute problems . Once your paper has been scheduled for online publication, the Nature press office will be in touch to confirm the details.

Acceptance of your manuscript is conditional on all authors' agreement with our publication policies (see www.nature.com/authors/policies/index.html). In particular your manuscript must not be published elsewhere and there must be no announcement of the work to any media outlet until the publication date (the day on which it is uploaded onto our web site).

Please note that *Nature Ecology & Evolution* is a Transformative Journal (TJ). Authors may publish their research with us through the traditional subscription access route or make their paper immediately open access through payment of an article-processing charge (APC). Authors will not be required to make a final decision about access to their article until it has been accepted. [Find out more about Transformative Journals](https://www.springernature.com/gp/open-research/transformative-journals)

Authors may need to take specific actions to achieve [compliance with funder and institutional open access mandates](https://www.springernature.com/gp/open-research/funding/policy-compliance-faqs). If your research is supported by a funder that requires immediate open access (e.g. according to [Plan S principles](https://www.springernature.com/gp/open-research/plan-s-compliance))

32then you should select the gold OA route, and we will direct you to the compliant route where possible. For authors selecting the subscription publication route, the journal's standard licensing terms will need to be accepted, including <https://www.nature.com/nature-portfolio/editorial-policies/self-archiving-and-license-to-publish>. Those licensing terms will supersede any other terms that the author or any third party may assert apply to any version of the manuscript.

We welcome the submission of potential cover material (including a short caption of around 40 words) related to your manuscript; suggestions should be sent to Nature Ecology & Evolution as electronic files (the image should be 300 dpi at 210 x 297 mm in either TIFF or JPEG format). Please note that such pictures should be selected more for their aesthetic appeal than for their scientific content, and that colour images work better than black and white or grayscale images. Please do not try to design a cover with the Nature Ecology & Evolution logo etc., and please do not submit composites of images related to your work. I am sure you will understand that we cannot make any promise as to whether any of your suggestions might be selected for the cover of the journal.

You can generate the link yourself when you receive your article DOI by entering it here: <http://authors.springernature.com/share>.

[REDACTED]

P.S. Click on the following link if you would like to recommend Nature Ecology & Evolution to your

33librarian <http://www.nature.com/subscriptions/recommend.html#forms>

** Visit the Springer Nature Editorial and Publishing website at http://editorial-jobs.springernature.com?utm_source=ejp_NEcoE_email&utm_medium=ejp_NEcoE_email&utm_campaign=ejp_NEcoE for more information about our career opportunities. If you have any questions please click [here](mailto:editorial.publishing.jobs@springernature.com).**